# Automated Detection of Visual Attribute Reliance with a Self-Reflective Agent

**Christy Li**[1]*  **Josep Lopez Camuñas**[2]  **Jake Thomas Touchet**[3]†  **Jacob Andreas**[1],
**Agata Lapedriza**[2,4],  **Antonio Torralba**[1],  **Tamar Rott Shaham**[1]*

[1]MIT CSAIL  [2]Universitat Oberta de Catalunya  [3]Louisiana Tech  [4]Northeastern University

## Abstract

When a vision model performs image recognition, which visual attributes drive its predictions? Detecting unintended reliance on specific visual features is critical for ensuring model robustness, preventing overfitting, and avoiding spurious correlations. We introduce an automated framework for detecting such dependencies in trained vision models. At the core of our method is a self-reflective agent that systematically generates and tests hypotheses about visual attributes that a model may rely on. This process is iterative: the agent refines its hypotheses based on experimental outcomes and uses a self-evaluation protocol to assess whether its findings accurately explain model behavior. When inconsistencies arise, the agent self-reflects over its findings and triggers a new cycle of experimentation. We evaluate our approach on a novel benchmark of 130 models designed to exhibit diverse visual attribute dependencies across 18 categories. Our results show that the agent's performance consistently improves with self-reflection, with a significant performance increase over non-reflective baselines. We further demonstrate that the agent identifies real-world visual attribute dependencies in state-of-the-art models, including CLIP's vision encoder and the YOLOv8 object detector.

## 1 Introduction

Computer vision models trained on large-scale datasets have achieved remarkable performance across a broad range of recognition tasks, often surpassing human accuracy on standard benchmarks [He et al., 2016, Dosovitskiy et al., 2020, Tan and Le, 2019, Russakovsky et al., 2015]. However, strong benchmark results can obscure underlying vulnerabilities. In particular, models may achieve high accuracy using prediction strategies that are non-robust or non-generalizable. These include relying on object-level characteristics such as pose or color [Geirhos et al., 2018], contextual cues like background scenery or co-occurring objects [Xiao et al., 2021, Alcorn et al., 2019], and demographic traits of human subjects [Wilson et al., 2019, Wang et al., 2019, Rosenfeld et al., 2018]. Such visual dependencies may result in overfitting, reduced generalization, and performance disparities in real-world usage [Hendrycks et al., 2021, Recht et al., 2019, Taori et al., 2020, Wiles et al., 2022].

Existing methods take various approaches to discover visual attributes that drive model predictions. These include saliency-based methods that highlight input regions associated with a prediction [Simonyan et al., 2013, Selvaraju et al., 2017, Kindermans et al., 2017], feature visualizations that map activations to human-interpretable patterns [Olah et al., 2017], and concept-based attribution methods that evaluate sensitivity to predefined semantic concepts [Kim et al., 2018, Ghorbani et al., 2019, Mu and Andreas, 2020]. While powerful for visualizing local behaviors, these approaches

---

*Address correspondence to: ckl@mit.edu, tamarott@mit.edu

†Work done while at MIT CSAIL

Website: https://christykl.github.io/saia-website/

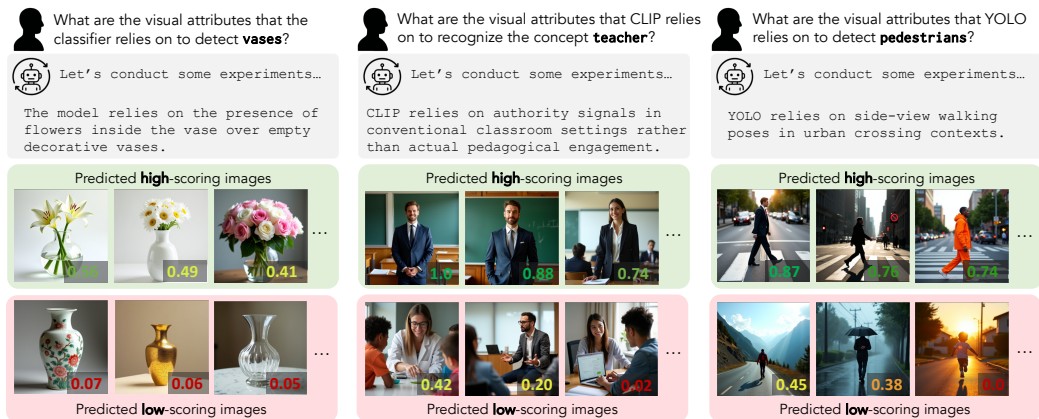

Figure 1: **Attribute Reliance Detection.** We use a self-reflective agent to produce natural-language descriptions of the visual attributes that a model relies on to recognize or detect a given concept. For each target concept (e.g., `vase`, `teacher`, or `pedestrians`), the agent first conducts hypothesis testing to reach a candidate description and then validates the description's predictiveness of actual model behavior through a self-evaluation protocol. The top row shows the agent's generated explanations. The bottom rows show images predicted to elicit high (green) or low (red) model responses, along with their actual model scores. Results are shown for different target concepts across an object recognition model with a controlled attribute reliance (left), CLIP (middle), and YOLOv8 (right).

often rely on manual inspection and assume access to a fixed set of predefined concepts, limiting their ability to scale to modern models with complex behaviors.

In this paper, we introduce a fully automated framework designed to detect visual attribute reliance in pretrained vision models. Given a pretrained model and a target visual concept (e.g., an image classifier selective for the object `vase`), our method identifies specific image features that systematically influence the model's predictions, even when these features fall outside the model's intended behavior (e.g., *the classifier relies on flowers to detect the vase*, Fig. 1). At the core of our approach is a self-reflective agent (implemented with a backbone multimodal LLM) that treats the task as a scientific discovery process. Rather than relying on a predefined set of candidate attributes, the agent autonomously formulates hypotheses about image features that the model might rely on, designs targeted tests, and updates its beliefs based on observed model behavior. In contrast to previous interpretability agents [Schwettmann et al., 2023, Rott Shaham et al., 2024], the self-reflective agent does not stop after generating an initial finding but rather actively evaluates how well it aligns the model's behavior on unseen test cases. Importantly, this evaluation is self-contained and does not require any ground-truth knowledge about the model's attribute dependencies. Instead, the agent generates two sets of test images: one where the candidate attribute is present (which is expected to elicit *high* prediction scores from the model) and one where the attribute is absent (which is expected to elicit *low* prediction scores). When discrepancies arise between the expected and actual model behaviors, the agent reflects on its assumptions, identifies gaps or inconsistencies in its current understanding, and initiates a new hypothesis-testing loop. We show that the agent's ability to reason about attribute reliance significantly improves with self-reflection rounds (see Sec. 5).

To quantitatively evaluate our method, we introduce a novel benchmark of 130 object recognition models, each constructed with a well-defined intended behavior and an explicitly injected attribute dependency. The benchmark spans 18 types of visual reliances, inspired by vulnerabilities known to exist in vision models [Dreyer et al., 2023, Geirhos et al., 2018, 2020, Buolamwini and Gebru, 2018, Xiao et al., 2021, Wang et al., 2019, Singh et al., 2020]. These include object-level attributes (e.g., color, material), contextual dependencies (e.g., background, co-occurring object state), and demographic associations. Together with an automated evaluation protocol, this benchmark provides a controlled environment for evaluating visual attribute reliance detection methods.

Our method is model-agnostic and can be applied to any vision model that assigns scores to input images. To demonstrate its versatility, we evaluate it on both our controlled benchmark and state-of-the-art pretrained models. Across a range of visual reliance types, our experiments show that our self-reflective agent consistently outperforms non-reflective baselines. Moreover, it successfully

uncovers previously unreported attribute dependencies in pretrained models (Fig. 1). For example, it identifies that the CLIP-ViT vision encoder [Radford et al., 2021] recognizes teachers based on classroom backgrounds and that YOLOv8 [Jocher et al., 2023], trained for object detection for autonomous driving, relies on the presence of crosswalks to detect pedestrians. These findings highlight the efficacy of our method as a scalable tool for detecting hidden dependencies in pretrained models deployed in real-world scenarios.

## 2 Related Work

**Revealing Visual Attribute Reliance in Vision Models.**    Prior work has explored methods to uncover the visual cues that drive model predictions. One common strategy is to manipulate input features to isolate model sensitivities, such as shape or spectral biases in classifiers [Gavrikov and Keuper, 2024], or attribute preferences in face recognition systems [Liang et al., 2023]. Other works rely on interpretability tools to identify potential dependencies—for example, extracting keywords from captions of misclassified images [Kim et al., 2024], or using feature visualizations to reveal facial attribute reliance [Teotia et al., 2022]. However, most of these methods target specific types of biases and rely on predefined concept sets or human inspection. In contrast, our framework introduces a unified, flexible approach that can detect a broad range of attribute reliances without prior assumptions about the relevant features.

**Interpretability and Automated Analysis.**    Initial work on interpretability automation produced textual descriptions of internal model features, using keywords [Bau et al., 2017], programs [Mu and Andreas, 2020], or natural language summaries [Hernandez et al., 2021, Bills et al., 2023, Gandelsman et al., 2023]. While informative, these descriptions are typically correlational, lacking behavioral validation [Huang et al., 2023, Schwettmann et al., 2023, Hausladen et al., 2024]. More recent work introduces agents that actively probe models. For instance, the Automated Interpretability Agent (AIA) [Schwettmann et al., 2023] used a language model to analyze black-box systems via a single pass over the input space. The Multimodal Automated Interpretability Agent (MAIA) [Rott Shaham et al., 2024] extended this approach by incorporating iterative experimentation and multimodal tools, enabling more detailed analysis of model internals. Our work builds on this direction by focusing specifically on discovering visual attribute reliances and introducing a self-reflection mechanism that allows the agent to revise faulty hypotheses based on experimental evidence, leading to more accurate and robust conclusions.

**Benchmarks for Visual Attribute Reliance Detection.**    Standardized benchmarks for evaluating visual attribute reliance remain limited. Prior evaluations often use models trained on datasets with known biases, such as WaterBirds [Wah et al., 2011] or CelebA [Liu et al., 2015], using label co-occurrence as a proxy for ground truth [Sagawa et al., 2020]. For generative settings, OpenBias [D'Incà et al., 2024] proposes biases via LLMs, generates images from biased prompts, and assesses reliance using VQA models. However, OpenBias is not applicable to predictive models and cannot conduct controlled interventions. We introduce a suite of 130 vision models with explicitly injected attribute reliances across 18 categories, providing fine-grained control over attribute type and strength. This allows rigorous, scalable evaluation of reliance detection methods in a predictive setting.

**Agent-Based Reasoning and Self-Reflective Systems.**    A growing body of work explores how agents can improve reasoning through reflection, feedback, and interaction. Methods like SELF-Refine [Madaan et al., 2023], Reflexion [Shinn et al., 2023], and ReAct [Yao et al., 2023] introduce multi-step loops in which agents revise their outputs via self-critique. Similarly, Du et al. [Du et al., 2023] show that multi-agent debate improves factual consistency and reasoning in language models. Our agent uses a task-specific self-evaluation protocol, enabling it to assess whether its conclusions align with actual model behavior. This integration of behavioral validation with self-reflection allows our agent to autonomously revise hypotheses and close the interpretability loop.

## 3 Self-Reflective Automated Interpretability Agent

Our framework is designed to automatically discover visual attributes that a pretrained model relies on to perform its task. Our approach consists of two main stages. (i) *Hypothesis-Testing stage*, in which an autonomous agent is provided with a subject model (e.g., an image classifier) and a target concept to explore (e.g., vase). The agent is tasked with discovering visual attributes in

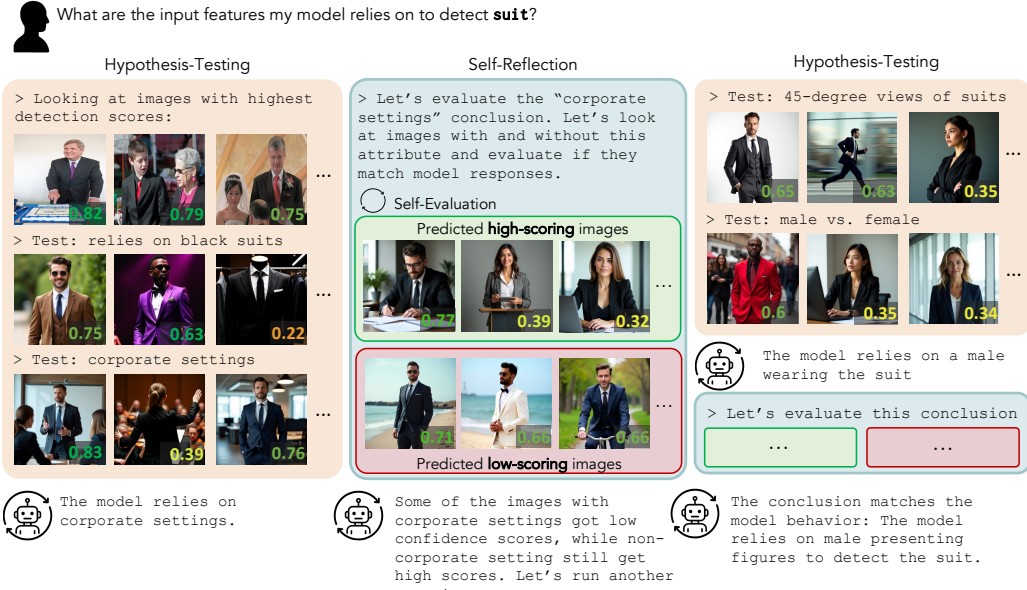

Figure 2: **Attribute reliance detection through hypothesis testing and self-reflection.** To discover features that drive model prediction, SAIA starts by formulating and testing a range of hypotheses. After reaching a conclusion (e.g., the model favors suit in corporate settings), it performs self-evaluation by testing the model responses on images with and without this feature. When inconsistencies between the conclusion and model behavior are observed (e.g., some non-corporate images yield high scores, while some corporate images yield lower scores), SAIA updates its prior beliefs according to these discrepancies and test alternative hypothesized explanations.

the input image that the subject model relies on to perform recognition tasks. The agent proposes candidate attributes that may influence the model's behavior, designs targeted experiments to test its hypotheses, and iteratively refines them based on observed results. This cycle continues until the agent converges to a stable explanation of the model's reliance. (ii) *Self-Reflection Stage*, in which the agent uses a self-evaluation tool to score its explanation. This is done by quantifying how well the agent's explanation matches the behavior of the model in new input images. If the explanation fails to generalize or reveals inconsistencies, the agent reflects on its prior explanation in light of the evaluation evidence and launches a new hypothesis-testing stage. Both stages are demonstrated in Fig. 2 and together form our **S**elf-reflective **A**utomated **I**nterpretability **A**gent (SAIA).

### 3.1 Hypothesis-Testing Stage

In this stage, SAIA iteratively refines hypotheses about the attribute sensitivities of the subject model. Inspired by MAIA [Rott Shaham et al., 2024], we design SAIA to operate in a scientific loop: it begins by proposing candidate attributes that the subject model might rely on, designs multiple experiments involving generating and editing images to test these hypotheses (e.g. edit an image with a suit to change its color), observes the resulting model's behavior (e.g. measure the subject model scores across these experiments) and updates its beliefs accordingly. This cycle continues until SAIA converges on an initial conclusion about the model's sensitivity to image features.

**Agent actions** SAIA interacts with the subject model through a set of predefined actions implemented as Python functions. These actions include: (i) querying the subject model with a given input image to observe its prediction score; (ii) retrieving the set of images that achieve the highest output responses from a fixed dataset, to identify inputs that strongly trigger the target concept; (iii) generating new images using a text-to-image model; (iv) editing existing images to manipulate specific attributes; (v) summarizing visual information across one or more images into text, to infer shared features; and (vi) displaying function which enables SAIA to log images, text, or other results in a notebook available throughout the experiment. SAIA designs experiments by composing multiple actions together through Python scripts. It then observes the experiment results—a combination of

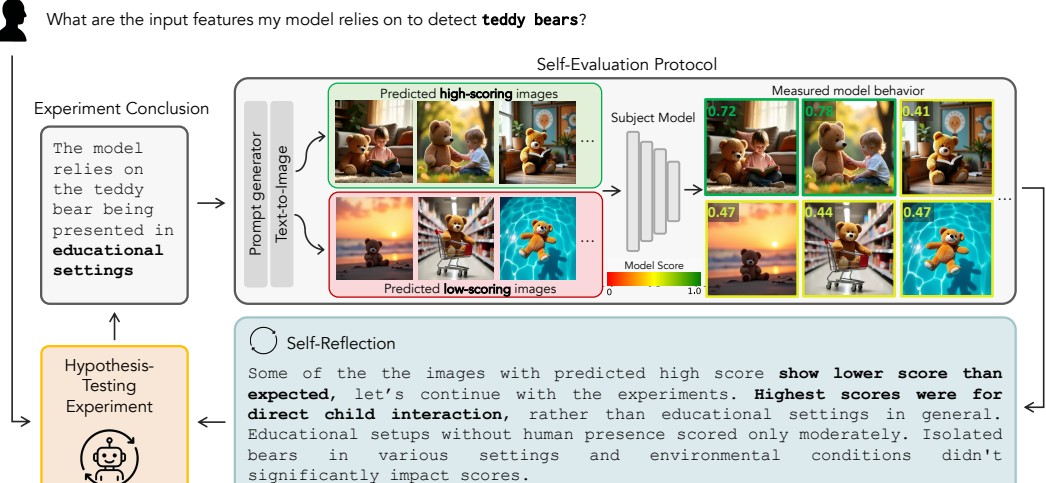

Figure 3: **Self-reflection stage.** SAIA initiates a hypothesis-testing experiment. After testing multiple candidate hypotheses (see Fig. 2), SAIA draws an inital conclusion (e.g., teddy bear are detected based on appearing in *educational settings*). SAIA then uses a self-evaluation protocol that generates synthetic images via a text-to-image model and computes the subject model's scores on these images. The self-evaluation protocol compares the predicted and actual model scores, triggering another round of hypothesis testing if results deviate from expectations. In this example, SAIA observes that the highest scores correlate with direct child interaction rather than generic educational settings, leading to refined future hypotheses.

text and images—and decides whether to continue with more experiments or to output a conclusion. We implement SAIA with a `Claude-Sonnet-3.5` backbone. Please refer to Appendix A for implementation details, full prompts, and API.

## 3.2 Self-Reflective Stage

Once the hypothesis testing stage is complete and SAIA reports its conclusion for the initial hypothesis-testing stage, we initiate a self-reflection stage. In this stage, the conclusion from the hypothesis testing stage is scored using a self-evaluation protocol. This protocol is completely unsupervised and does not require any ground-truth labels or external information. Instead, it measures how well the detected reliance matches the actual behavior of the subject model. If SAIA's detected reliance sufficiently matches the model's behavior, it terminates the experiments and returns the current conclusion. Otherwise, if inconsistencies between SAIA's conclusion and the model behavior are found, the information collected from the self-evaluation stage is returned to SAIA, which reflects over its previous conclusion and initiates another hypothesis-testing round. This process is demonstrated in Fig. 3.

**Self-Evaluation protocol** Self-evaluation serves two key purposes: (i) to assess whether the current explanation matches the model's behavior, and (ii) to guide further experimentation if it does not. The process begins by querying a separate language model instance (we use `Claude-Sonnet-3.5`) to generate two diverse sets of image prompts, that according to SAIA's description are predicted to elicit high and low detection scores, respectively. The first set, termed the "predicted high-scoring images", contains instances of images with the target concept (e.g., `teddy bear`) along with the detected reliance attribute (e.g., educational settings) that SAIA detected the model to be selective for. The second set, the "predicted low-scoring images", contains instances of the target concept, but with the absence of the detected attribute. This prompt generation strategy emphasizes attribute-controlled diversity, where each prompt in the high or low-scoring group keeps the core attribute constant while allowing other visual factors to vary. These prompts are then used as inputs to a text-to-image model (we use `Flux.1-dev` Labs [2024]) that generates the corresponding images, which are then fed to the subject model, and the output scores are recorded. If SAIA's explanation is accurate, the model should exhibit systematically higher scores on the "predicted high-scoring" images and lower scores on the "predicted low-scoring" set.

Behavior-matching protocols of this kind have been shown to be effective in other evaluation settings, particularly for the task of producing textual labels of neurons' behavior in pretrained models [Rott Shaham et al., 2024, Kopf et al., 2024, Huang et al., 2023]. In domains where ground-truth explanations are unavailable, they provide a way to validate hypotheses through behavioral consistency. We repurpose this evaluation method as a basis for self-reflection: SAIA uses it to validate its own conclusions, determine whether further experimentation is necessary, and reflect on its own findings based on measured model behavior.

**Agent self-reflection** After observing the model's responses to the "predicted high-scoring" and "predicted low-scoring" images, SAIA reflects on whether the results align with its expectations. If the mean scores between the two groups are not sufficiently separated based on an empirically set threshold (i.e. the conclusion is not sufficiently discriminative), SAIA may decide that its current conclusion is incomplete or inaccurate. It then analyzes which visual attributes within the generated images might explain these discrepancies. In doing so, SAIA updates its hypothesis — either by narrowing the original explanation (e.g., refining "educational settings" to "child interaction") or by generating alternative hypotheses altogether. This reflective process closes the experimental loop and allows SAIA to reinitiate the hypothesis-testing stage with better-informed guidance.

In practice, we cap the total number of agent rounds (hypothesis-testing followed by self-reflection) to 10. If no hypothesis meets the self-evaluation threshold by that point, SAIA returns the hypothesis that achieved the best alignment between predicted and actual model behavior, typically the most recent one. Empirically, SAIA converges well before reaching this cap — in most runs, it stops after just 2-4 rounds. This behavior is visualized in Figure 5a, which shows that the predictiveness score (see definition in Sec. 5.1) generally improves monotonically across rounds, providing a natural convergence signal. Please refer to Appendix A.2 for the full self-reflection instructions.

## 4 A Benchmark of Models with Controlled Attribute Reliance

To evaluate the capabilities of SAIA, we constructed a benchmark of 130 unique object recognition models that exhibit 18 diverse types of visual attribute reliance. All simulated behaviors are inspired by known vulnerabilities of vision models [Dreyer et al., 2023, Geirhos et al., 2018, 2020, Buolamwini and Gebru, 2018, Xiao et al., 2021, Wang et al., 2019, Singh et al., 2020], and mimic spurious correlations between the target object and image attributes such as object color, background context, co-occurring object state, or demographic cues. To assess the generalizability of our method, the benchmark also includes a subset of models with *counterfactual* attribute reliance that are intentionally rare or unnatural in real-world pretrained models (e.g., a suit detector responds more strongly when a women wear the suit). Each benchmark model includes an input parameter that controls the strength of the injected reliance, allowing for precise control over model behavior. Importantly, because these models are explicitly engineered with a known intended behavior, they serve as a controlled testbed for evaluating and comparing feature reliance detection methods.

### 4.1 Simulating attribute dependencies

Figure 4 illustrates a simulated attribute reliance scenario. Given a target object class $t$ (e.g. bird) and an intended injected attribute reliance $i$ (e.g. setting; beach), we simulate a model $\mathcal{C}_{t,i}$ that detects $t$ under the condition $i$. Each benchmark model is composed of two components; an object detector $\mathcal{O}_t$ and an attribute condition detector $\mathcal{A}_i$, which modulates the output of $\mathcal{O}_t$ based on the presence or absence of the specified attribute. To compute the final output of the model $\mathcal{C}_{t,i}$ on an input image img, we first pass the image through the object detector $\mathcal{O}_t$. If the target object class is not detected, the model returns a low random baseline score. If the object is detected, the image is then evaluated by the attribute condition detector $\mathcal{A}_i$. If the attribute condition is satisfied, the original subject model score $\mathcal{O}_t(\text{img})$ is returned, simulating full model response. Otherwise, the subject model score is discounted by a multiplication factor of $\alpha$, simulating attenuated confidence due to the missing attribute. The scalar $\alpha \in [0, 1]$ controls the magnitude of the injected reliance: higher values of $\alpha$ simulate stronger reliance on the attribute. Please refer to Appendix B.2 for empirical evaluation of reliance magnitude as a function of $\alpha$. In all the benchmark models, we use Grounding DINO [Liu et al., 2023] as the object detector $\mathcal{O}_t$ and SigLIP [Zhai et al., 2023] as the attribute condition detector $\mathcal{A}_i$, which in practice is guided by a textual description of the injected attribute condition $i$. For demographic attribute dependencies, FairFace [Karkkainen and Joo, 2021] is used for $\mathcal{A}_i$ instead

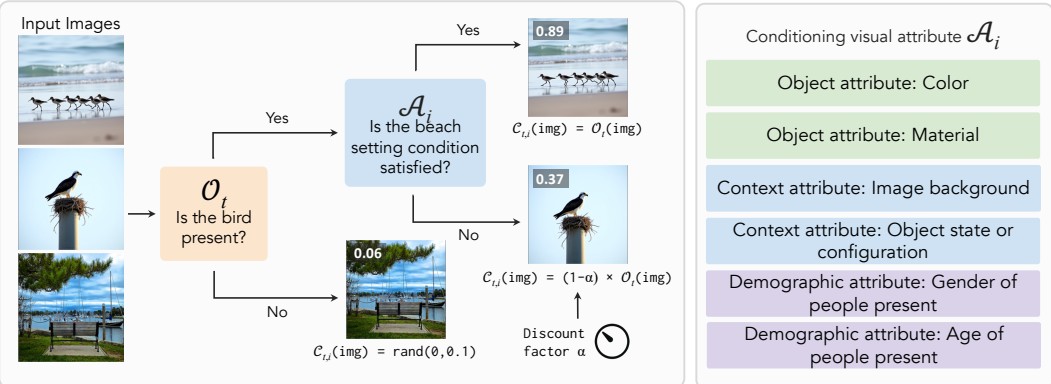

Figure 4: **Simulating visual feature reliance.** We simulate feature reliance by modulating object recognition scores based on the presence of specific visual attributes (e.g., a `bird` detector that relies on the presence of `beach background`). Given an input image and object category $t$, $\mathcal{O}_t$ produces a object recognition score for object presence. If the object is not detected, a low random score is assigned as the final model score of the image. If the object is detected, we simulate an attribute dependency (e.g., presence of a "beach background" for bird detection) through the procedure described in Sec. 4.1. If the condition is satisfied, the final model score equals $\mathcal{O}_t(\text{img})$ recognition score. Otherwise, the score is discounted by a factor $\alpha$ to represent the model's weaker response in the case that the attribute condition is not met.

of SigLIP (see implementation details below). Notably, this model composition approach is highly flexible—one could engineer any object-condition pairing to construct an object detection model with a desired attribute reliance.

## 4.2 Attribute Condition Categories

We categorize the attribute conditions used to inject reliance into four groups: object attributes, context attributes, demographic attributes, and counterfactual demographic attributes. These categories reflect different types of visual dependencies observed (or intentionally constructed) in our benchmark models, and guide the choice of attribute detector $\mathcal{A}$ used in each case. Please see the full list of constructed models in Appendix B.

**Object attributes** These attribute dependencies relate to visual properties of the object itself. We include reliance on object *color* and *material*, using SigLIP as $\mathcal{A}$ for zero-shot classification of object-specific attributes (e.g. SigLIP is guided with the prompt `a red bus` to inject a color reliance to a bus detector). A color-reliant system returns the full score from $\mathcal{O}_t$ only if the object has a specific color, otherwise the response is discounted; similarly, a material-reliant model will have a full response only if the object is of the intended material (e.g., `vases made of ceramic`).

**Context attributes** These dependencies reflect properties of the object's surrounding context. We simulate reliance on the specific *setting* of the object (e.g. `keyboard` only if it is being typed) and *object background* (e.g. `car` only if it is in an urban environment). Here as well, we use SigLIP-guided text for detecting the intended attribute.

**Demographic attributes** These dependencies are based on the age or gender of people interacting with the target object. We use FairFace as $\mathcal{A}$ to detect demographic attributes and construct systems relying on these (e.g., an apron detector that relies on the apron to be worn by women, and a glasses detector that relies on the glasses to be worn by older individuals).

**Counterfactual demographic attributes** To test whether SAIA can discover atypical or out-of-distribution dependencies, we include models with *counterfactual* demographic reliance (e.g., an apron detector that activates only when worn by men, or a glasses detector that prefers younger wearers), which rarely co-occurs in real-world data. These systems allow us to test whether SAIA can detect unexpected or counterintuitive reliance patterns that do not follow natural co-occurrence statistics. This distinction allows us to assess both SAIA's ability to uncover realistic demographic biases and its robustness to rare or previously unknown dependencies.

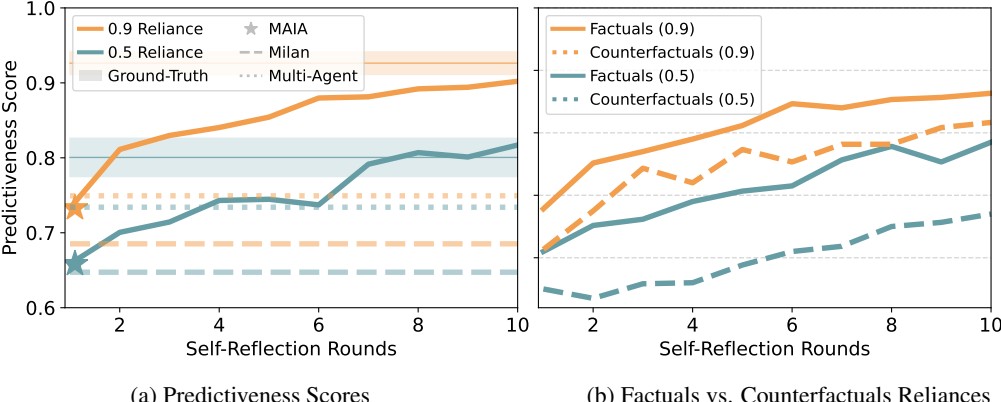

(a) Predictiveness Scores       (b) Factuals vs. Counterfactuals Reliances

Figure 5: **Predictiveness Score over Self-Reflection Rounds.** **(a)** We plot the average predictiveness score over all models in the benchmark for a harsh reliance magnitude of $\alpha = 0.9$, as well as a softer reliance magnitude of $\alpha = 0.5$. We see a steady increase in the predictiveness scores of SAIA's conclusions over rounds for both discount factors, approaching their respective theoretical upper bounds as given by the ground truth baseline. As expected, the scores for softer reliance models ($\alpha = 0.5$) are consistently lower than those of the stronger reliance models ($\alpha = 0.9$), illustrating that subtler attribute reliances are more challenging to detect. SAIA outperforms all nonreflective baselines (MAIA, Milan, and Multi-Agent) by a significant margin for both reliance magnitude values. **(b)** We compare the predictiveness scores of SAIA's reliance descriptions over factual models with more intuitive demographic attribute reliances against counterfactual reliances on object-demographic associations that are not commonly observed. Although SAIA's descriptions of the counterfactual models achieve lower predictiveness scores, the performance still reliably improves over increased rounds of self-reflection for both $\alpha$ settings.

## 5 Experiments

We evaluate the performance of SAIA on both our synthetic benchmark models and on pretrained vision models widely used in practical settings. Examples of attribute reliance discovered by SAIA, as well as evaluation results, are shown in Figures 1, 2, 3, and 7.

### 5.1 Evaluation protocol

We quantitatively evaluate the accuracy of the detected attribute reliances generated by SAIA and compare its performance to four different baselines.

**Predictiveness score** Following Kopf et al. [2024], Schwettmann et al. [2023], we quantify how well a candidate reliance description matches model behavior. Similar to the self-evaluation score, given a candidate explanation, we start by generating 10 synthetic images that are expected to elicit *high* model scores and 10 that are expected to elicit *low* scores. We then pass these images through the model and record its actual responses. Each image is assigned a binary prediction label (high or low predicted response), and we threshold the model's scores to obtain a binary outcome (high or low measured response). The predictiveness score is computed as the proportion of images where the predicted label matches the model's actual binary output. This reflects how well the explanation predicts individual model responses.

**LLM as a Judge** We use a language model as a judge [Zheng et al., 2023] in a two-alternative forced choice (2AFC) setting. Given a ground-truth explanation of a benchmark model and two candidate explanations, one produced by SAIA and one from a baseline, the LLM judge is asked to choose which candidate better matches the ground-truth description. For each description pair, we repeat the test 10 times, and report the average preference rate for SAIA's descriptions.

**Baselines** We compare SAIA against the following alternatives: (i) *Milan-style reliance detection*: Following Milan [Hernandez et al., 2021], this approach avoids iterative experimentation and detects the reliance based on a precomputed set of image exemplars that maximize the model's scores.

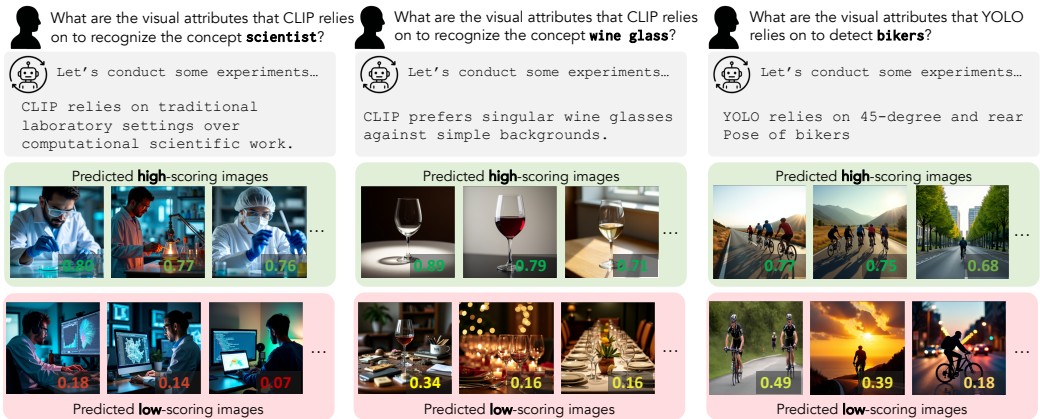

Figure 7: **Detected feature reliance in CLIP and YOLO.** SAIA identifies visual attribute dependencies in state-of-the-art pretrained models that have not been previously documented. For each concept (e.g., *scientist*, *wine glass*, *biker*), SAIA infers an attribute reliance through a natural-language explanation and tests it by comparing predicted high and low-scoring images (scores are normalized for clarity). The examples reveal that CLIP-ViT relies on traditional laboratory settings to recognize `scientists`, while YOLOv8 favors 45-degree and rear views for detecting `bikers`.

(ii) *MAIA-style agent*: Based on Rott Shaham et al. [2024], this method performs hypothesis testing but does not engage in self-reflection to revise its explanation. For a fair comparison, we equip MAIA with the same set of tools and backbone model that our method uses. (iii) *Multi-agent ensemble*: We run 10 independent MAIA-style, non-self-reflective agents and select the explanation with the highest predictiveness score. This tests whether repeated sampling alone can match the performance gains from self-reflection. (iv) *Ground-truth descriptions*: For benchmark models, we include ideal natural language descriptions of the injected reliance as an upper bound on performance.

| $\alpha$ | SAIA vs. Milan | SAIA vs. MAIA | SAIA vs. Multiagent |
|---|---|---|---|
| 0.5 | $0.54 \pm 0.02$ | $0.56 \pm 0.03$ | $0.53 \pm 0.03$ |
| 0.9 | $0.59 \pm 0.02$ | $0.6 \pm 0.02$ | $0.62 \pm 0.03$ |

Figure 6: **2AFC evaluation.** We use a language model (GPT-4) as a judge. Given a ground-truth description, the LLM compares two candidate explanations: one generated by SAIA and one from a competitive baseline (MAIA, Milan, or Multiagent). The LLM selects the candidate it finds most semantically similar to the ground truth. We report average preference rate for SAIA in the table.

## 5.2 Evaluating Benchmark Models

**Self-reflection enhances reliance detection**
As showed in Fig. 5a, predictiveness scores steadily improve over the course of self-reflection rounds, suggesting that SAIA's explanations become increasingly aligned with the model's actual behavior. Notably, performance exceeds the MILAN one-shot baseline, a MAIA-style agent, and the non-reflective multiagent system, indicating that self-reflection offers a distinct advantage.

**Robust performance across different degrees of model reliance**  Figure 5a shows consistent performance gains for both strong ($\alpha = 0.9$) and weak ($\alpha = 0.5$) reliance settings, demonstrating that SAIA is effective across a range of dependency strengths. While stronger dependencies lead to higher absolute scores, the relative improvement from self-reflection remains significant even in more ambiguous scenarios. The same trend is seen in the 2AFC test (Fig. 6), where the final conclusion from SAIA is more frequently preferred over the baselines for the stronger reliance.

**SAIA discovers counterfactual feature reliances**  In addition to recovering realistic dependencies, SAIA successfully identifies *counterfactual* attribute reliances (Fig. 5b). This indicates that SAIA is capable of discovering surprising or non-intuitive patterns of reliance, rather than simply mirroring familiar dataset biases.

## 5.3 Revealing attribute reliance in pretrained vision models

We deploy SAIA to detect attribute reliances in two pre-trained vision models: the CLIP-ViT image encoder [Radford et al., 2021] trained to align image and text representations, and the YOLOv8 model [Jocher et al., 2023] trained for object detection in autonomous driving settings. With CLIP, we perform object recognition by measuring the cosine similarity of the image with a target prompt (e.g., *"A picture of a scientist"*). For YOLOv8 we measure the detection score of the target object class. Figures 1 and 7 show that SAIA can generate natural-language descriptions of attribute reliance in various contexts. The generated descriptions are shown to be *predictive* of model behavior, as model scores increase when the reliance is satisfied and decrease when it is absent. Surprisingly, SAIA reveals dependencies that were never observed before, such as the reliance of clip on traditional laboratory settings when detecting `scientist`, and YOLOv8 dependency on `bikers`' poses. We note that SAIA's goal is to surface such dependencies rather than to assess their desirability or harm, allowing practitioners to make informed judgments based on specific downstream use cases.

## 5.4 Revealing compositional visual reliance

Vision models can rely on combinations of multiple attributes to detect certain concepts. To evaluate SAIA's performance in such settings, we created ten additional synthetic benchmark models that each exhibit two attribute reliances. Five models depend on the simultaneous presence of *both* attributes (e.g., a bench detector that relies on benches that are *wooden* AND in *beach settings*), while the other five exhibit high confidence if either attribute is present (e.g., *wooden* benches OR benches in beach settings). Model details are provided in Table 6 of the Appendix.

For models requiring both attributes simultaneously (AND logic), SAIA successfully identified both dependencies in $80\%$ of cases, whereas MAIA (the non-self-reflective baseline) failed to detect both attributes in any case. This performance gap stems from SAIA's self-evaluation protocol, which rigorously tests multi-attribute hypotheses by constructing positive exemplars containing all relevant attributes and negative exemplars containing none or only a subset of candidate attributes. For models relying on at least one of two attributes (OR logic), both methods struggle more significantly. SAIA recovered both dependencies in only $20\%$ of synthetic models, while MAIA again failed to detect both dependencies. This difficulty likely arises from terminating the process once one attribute is found. Future extensions could address this limitation with a hierarchical or tree-structured experimental design that explicitly enforces multi-reliance exploration. See Appendix C.2 for further results and analysis of these experiments.

Overall, these findings demonstrate that self-reflection enhances SAIA's robustness and compositional reasoning, leading to more reliable explanations even when multiple visual dependencies are involved. Beyond multi-attribute detection, the self-reflective stage also improves performance across broader experimental settings. One limitation of agentic interpretability methods is that relying on external tools introduces errors stemming from tool inconsistencies or biases. A key distinction in our framework is the inclusion of a self-reflection loop, which help to mitigate such artifacts. Please refer to Appendix C.3 for experimental details and analysis.

## 6 Conclusion

We introduced SAIA, a self-reflective agent for discovering attribute reliance in pretrained vision models. Treating interpretability as a scientific discovery process, SAIA autonomously generates hypotheses, tests them through synthetic image generation and editing, and refines its conclusions via a self-evaluation loop. Our experiments on controlled benchmarks and real-world models as CLIP and YOLOv8 show that SAIA outperforms non-reflective baselines and uncovers both expected and previously unreported attribute dependencies. While the method provides a scalable framework for behavioral model auditing, its performance depends on the quality of generated images, which can limit robustness for rare or out-of-distribution concepts. Moreover, the agent's backbone model inherits known priors and biases that may constrain the hypothesis space and introduce confirmation bias—likely explaining the reduced performance on counterfactual demographic models, though self-reflection still improves predictive reliability. Despite these challenges, self-reflective agents mark a promising step toward transparent and automated model auditing, and their effectiveness will continue to grow with advances in image generation and agent reasoning.

## Acknowledgments

We are grateful for the support of ARL grant #W911NF-24-2-0069, MIT-IBM Watson AI Lab grant #W1771646, Hyundai Motor Company, ONR MURI grant #033697-00007, and PID2022-138721NB-I00, funded by MCIN/AEI/10.13039/501100011033 and FEDER, EU. This research is (partially) supported by the DARPA AIQ program through the DARPA CMO contract number HR00112520025.

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

# Appendix

## A SAIA Implementation

### A.1 Hypothesis loop instructions

You will get an access to a classifier in the final layer of a deep neural network that is sensitive to the presence of <concept> within photo-realistic images. Your overall task is to reveal biases in the classifier's behavior. That is, your task is to find specific features for which the classifier tends to have relatively higher confidence scores, but only covers partial aspect of the <concept> class, or specific features for which the classifier tends to have relatively lower confidence scores although <concept> is still present.

Try to identify and explain the cause of biases in the network. These biases could reflect real-world biases or they could be spurious. Some bias categories include:

1. Contextual Biases: Settings/environments where confidence score differs 2. Demographic Biases: Variations across race, gender, age, etc. 3. Technical Biases: Lighting, angle, scale, or quality dependencies 4. Spurious Biases: Correlations between classes and spurious features

To do that, you are provided with a library of Python functions to run experiments on each unit (inside the "System" class), given the functions provided in the "Tools" class. Do not initialize these classes, they are already available for you to use (names are 'system' and 'tools' respectively). Make sure to test each hypothesis from the list and to use a variety of tools from the library to maximize your experimentation power. At each experiment step, write Python code that will conduct your experiment on the model, using the following format:

```
[CODE]: # write Python code using your knowledge of Python and
the System and Tools # class for running experiments.  You must
display values if you want # to see them.  They will not be displayed
automatically.
```

Include only a single instance of experiment implementation at each step. Your code will be executed and the results will be displayed. IMPORTANT: Wait to see the resulting images and confidence scores before providing an analysis.

Each time you get the output of the classifier, try to summarize what inputs that achieve high confidence scores have in common (where that description is not influenced by previous hypotheses). Then, write multiple hypotheses that could explain the biases of the classifier. For example, these hypotheses could list multiple features that the classifier is more or less sensitive to.

After each experiment, wait to observe the outputs of the classifier and analyze the results from each experiment. [ANALYSIS]: Let's break this down step by step. The experiments imply that...

Based on these conclusions, write a list of initial hypotheses about the classifier biases in the format: [HYPOTHESIS LIST]: Hypothesis$_1$: <hypothesis$_1$> ... Hypothesis$_n$: <hypothesis$_n$>

Now, write additional experiments to test them. Test the effects of both local and global differences in images using the different tools in the library. If you are unsure about the results of the previous experiment, you can also rerun it, or rerun a modified version of it with additional tools.

Use the following format: [CODE]: ##conduct additional experiments using the provided python library to test *ALL* the hypotheses. Test different and specific aspects of each hypothesis using all of the tools in the library. Write code to run the experiment in the same format provided above. Include only a single instance of experiment implementation. Wait for the code to be executed and the results to be prompted to you before continuing to the analysis step. [ANALYSIS]: ## systematically draw conclusions from each tested hypothesis. HYPOTHESIS LIST]: ## update your hypothesis list according to the image content and related confidence scores.

Continue running experiments until you prove or disprove all of your hypotheses. Only when you are confident in your hypothesis after proving it in multiple experiments and no additional experiments need to be executed, output your final description of the classifier bias in the following format. Output your final description of the classifier bias ONLY ONCE, when no more [CODE] is provided for additional experiments. DO NOT write any code in the round you

want to output your final bias description and bias label: [BIAS DESCRIPTION]: <description of bias>

In addition, output a one-sentence summary of the bias description in the following format: [BIAS DESCRIPTION]: <succinct one-sentence summary of bias description>

Examples ——— # Example 1 [BIAS LABEL]: The classifier should classify dogs, but it has a bias to prefer white dogs. =======

# Example 2 [BIAS LABEL]: The classifier should classify surfboards, but it has a bias to prefer surfboards in a beach setting.

Remember: - Document all images and their confidence scores by using the display function - Wait for experiments to execute and results to be shown to you before drawing any conclusions or revising any hypotheses - Verify findings with multiple approaches - DO NOT use any functions other than the ones provided

You should maintain a systematic approach while being creative in hypothesis generation and testing. Each experiment should build on previous findings while maintaining methodological rigor.

## A.2  Self-reflection instructions

You will get an access to a classifier in the final layer of a deep neural network that is sensitive to the presence of 0 within photo-realistic images. Your overall task is to reveal biases in the classifier's behavior. That is, your task is to find specific features for which the classifier tends to have relatively higher confidence scores, but only covers partial aspect of the 0 class, or specific features for which the classifier tends to have relatively lower confidence scores although 0 is still present.

Try to identify and explain the cause of biases in the network. These biases could reflect real-world biases or they could be spurious. Some bias categories include:

1. Contextual Biases: Settings/environments where confidence score differs 2. Demographic Biases: Variations across race, gender, age, etc. 3. Technical Biases: Lighting, angle, scale, or quality dependencies 4. Spurious Biases: Correlations between classes and spurious features

A fellow scientist has proposed the following hypothesis about this classifier's bias:

HYPOTHESIS: "BIAS"

We have conducted experiments on three sets of relevant images to help you assess this hypothesis:

1. [DATASET EXEMPLARS]: 10 real-world images achieving highest classifier confidence score - These represent the classifier's "natural" high-confidence score patterns - Each includes confidence score and classification (HIGH/MODERATE/LOW)

2. [POSITIVE EXAMPLES]: 10 synthetic images designed to test the bias hypothesis - Expected to trigger high confidence score based on the hypothesis - Includes generation prompt, confidence score, and classification (HIGH/MODERATE/LOW)

3. [NEGATIVE EXAMPLES]: 10 synthetic images as hypothesis controls - Expected to trigger low confidence score based on the hypothesis - Includes generation prompt, confidence score, and classification (HIGH/MODERATE/LOW)

ANALYSIS GUIDELINES:

1. High Confidence Score Pattern Analysis: - Group all high-activating images together, regardless of which set they're from - Identify common visual elements across these high-confidence score cases - Extract shared prompt elements that consistently produce high confidence scores - Note cases where similar prompts produced different confidence score levels

2. Cross-Reference with Exemplars: - Compare high-activating synthetic images to exemplar patterns - Identify prompt elements that successfully replicate exemplar features - Note any systematic differences between synthetic and exemplar cases

3. Confidence Assessment: - Evaluate consistency of prompt-to-confidence score relationships - Identify which prompt patterns reliably produce high confidence scores - Note any context-dependent variations in prompt effectiveness

Based on these experimental results and analysis, generate a comprehensive list of possible biases to explore with further experimentation. Present your hypotheses in the following format:

[HYPOTHESIS LIST]: Hypothesis$_1$: <hypothesis$_1$> ... Hypothesis$_n$: <hypothesis$_n$>

Strong hypotheses should explain both intended and unintended high-confidence score patterns while identifying reliable prompt elements that trigger the classifier.

## A.3 Image generation for self-evaluation

You will get access to a classifier in the final layer of a deep neural network that detects the presence of 0 within photo-realistic images for autonomous driving. Your task is to validate a hypothesis about the classifier's biases. A researcher has concluded:

HYPOTHESIS: <BIAS>

Generate two sets of image prompts: 1. 10 prompts for images containing 0 that should trigger HIGH classifier confidence score based on the hypothesis 2. 10 prompts for images containing 0 that should trigger LOW classifier confidence score based on the hypothesis

If the hypothesis suggests multiple bias scenarios, distribute the prompts to cover all cases. Make sure each image only tests ONE bias scenario at a time so different possible sources of bias do not confound.

Return your results in this exact format:

<POSITIVE PROMPTS> 1. [First high-confidence score prompt] 2. [Second high-confidence score prompt] ... 10. [Tenth high-confidence score prompt] </POSITIVE PROMPTS>

<NEGATIVE PROMPTS> 1. [First low-confidence score prompt] 2. [Second low-confidence score prompt] ... 10. [Tenth low-confidence score prompt] </NEGATIVE PROMPTS>

Remember to start the positive examples with the opening tag <POSITIVE PROMPTS> and end the positive examples with closing tag </POSITIVE PROMPTS>. Remember to start the negative examples with opening tag <NEGATIVE PROMPTS> and end the negative examples with closing tag </NEGATIVE PROMPTS>.

## A.4 SAIA API and implementation details

SAIA's API is based on that of MAIA Rott Shaham et al. [2024] with a few important modifications: (i) SAIA is able to define experiments in free-form code blocks, where initially MAIA had to define all its code within a function (execute_command) that was then executed by the codebase. This allows SAIA to both write multiple blocks of code per experiment and to access variables defined in previous experiments. (ii) SAIA is now able to log and display any text/images generated during its experiments in the format of its choosing with a single, flexible function (display), whereas MAIA had to rely on individual tools to display their own results in predetermined formats. (iii) SAIA can use more recent VLLMs, particularly Claude-3.5-sonnet, where the original codebase used gpt-4-vision-preview.

## A.5 Agent tools

To support experimentation and interpretability analysis, we provide a `Tools` class with utilities for:

- **Text-to-image generation** from prompts via pretrained diffusion models.
- **Prompt-based image editing** to create controlled counterfactuals.
- **Image summarization** that identifies common semantic or non-semantic features across a set of images.
- **Region-based image descriptions** that generate textual descriptions of highlighted activation regions.
- **Exemplar retrieval** for a given classifier unit, returning representative images and their scores.

## A.6 Supported backbone models.

SAIA ships with a small self-contained toolkit that lets us run end-to-end vision experiments (image generation, editing, and logging) through a single, uniform interface.

- **Text-to-Image Generation**:
  - Flux Image Generator Labs [2024]
  - DALL·E 3 OpenAI [2023] (OpenAI API) (used for the evaluation)
- **Image Editing**:
  - InstructDiffusion (Stable Diffusion variant with instruction tuning) Geng et al. [2023]
- **Image Description and Summarization**:
  - GPT-4o OpenAI [2024], used via API to describe image regions and summarize visual commonalities across images.

## A.7 Interface and Logging.

All generated or edited images are stored in Base64 format for transmission and display. The framework logs each experiment (prompt, image, activation, description) and supports export as an interactive HTML report for reproducibility.

Overall, the toolkit enables SAIA to generate or edit images for hypothesis testing, score them different models, analyse the outcomes, and package the entire run into a report, making each experiment swift, scalable, and fully reproducible. For a comprehensive overview of hardware requirements, see Table 1.

## A.8 Resources

All our experiments were conducted on a single NVIDIA RTX 3090 (24 GB) GPU. SAIA's backbone (Claude-3.5-sonnet) was used through Anthropic API, and the prompt generator for self-reflection (GPT4o) and the evaluator modern in the 2AFC experiment (GPT4) were accessed through OpenAI API. An experiment with 10 rounds of hypothesis testing followed by self-reflection costs approximately $3 and takes about 10-20 minutes per round. Note that most experiments are concluded before 10 rounds, so this is an upper bound.

| Model (Inference) | Peak VRAM↓ (GB) | #Params↓ (M) |
|---|---|---|
| SAM ViT–H | ∼7.0 | 632 |
| Grounding DINO Swin–T | ∼0.45 | 174 |
| SigLIP So400m (P14/384) | ∼2.1 | 878 |
| CLIP ViT–L/14 (OpenAI) | ∼2.04 | 428 |
| FairFace ResNet-34 | ∼0.06 | 21.8 |
| YOLOv8-m (Ultralytics) | ∼0.07 | 25.9 |
| Stable Diffusion 3.5 Medium (FP16) | ∼9–10 | 2 500 |
| FLUX.dev (12B, 4-bit + offload) | ∼10-11 | 12 000 |
| InstructPix2Pix (SD-1.5 base) | ∼6-7 | 890 |
| Instruction Diffusion | ∼10 | 1 000 |
| RetinaFace MobileNetV3 | ∼0.02 | 1.7 |
| Average Experiment | ∼19.5-20.5 | – |

Table 1: Peak GPU memory and parameter scale of all models used at inference time.

## A.9 API prompt

```python
class System:
    """
    A Python class containing the vision model and the specific classifier to
        interact with.

    Attributes
    ----------
    classifier_num : int
        The unit number of the classifier.
    layer : string
        The name of the layer where the classifier is located.
    model_name : string
        The name of the vision model.
    model : nn.Module
        The loaded PyTorch model.

    Methods
    -------
    call_classifier(image_list: List[torch.Tensor]) -> Tuple[List[int], List[str]]
        Returns the classifier confidence score for each image in the input
            image_list as well as the original image (encoded into a Base64 string)
            .
    """

    def __init__(self, classifier_num: int, layer: str, model_name: str, device:
        str):
        """
        Initializes a classifier object by specifying its number and layer location
            and the vision model that the classifier belongs to.
        Parameters
        -------
        classifier_num : int
            The unit number of the classifier.
        layer : str
            The name of the layer where the classifier is located.
        model_name : str
            The name of the vision model that the classifier is part of.
        device : str
            The computational device ('cpu' or 'cuda').
        """
        self.classifier_num = classifier_num
        self.layer = layer
        self.device = torch.device(f"cuda:{device}" if torch.cuda.is_available()
            else "cpu")

    def call_classifier(self, image_list: List[torch.Tensor]) -> Tuple[List[float],
        List[str]]:
        """
        The function returns the classifier's maximum confidence score (in float
            format) for each of the images in the list as well as the original
            image (encoded into a Base64 string).

        Parameters
        ----------
        image_list : List[torch.Tensor]
            The input image

        Returns
        -------
        Tuple[List[int], List[str]]
            For each image in image_list returns the confidence score of the
                classifier on that image, and the original image encoded into a
                Base64 string.
```

```python
        Examples
        --------
        >>> # test the confidence score of the classifier for the prompt "a dog
            standing on the grass"
        >>> prompt = ["a dog standing on the grass"]
        >>> image = tools.text2image(prompt)
        >>> score_list, image_list = system.call_classifier(image)
        >>> for score, image in zip(score_list, image_list):
        >>>     tools.display(image, f"Confidence Score: {score}")
        >>>
        >>> # test the confidence score of the classifier for the prompt "a dog
            standing on the grass" and maintain robustness to noise
        >>> prompts = ["a dog standing on the grass"]*5
        >>> images = tools.text2image(prompts)
        >>> score_list, image_list = system.call_classifier(images)
        >>> tools.display(image_list[0], f"Confidence Score: {statistics.mean(
            score_list)}")
        >>>
        >>> # test the confidence score of the classifier for the prompt "a
            landscape with a tree and river"
        >>> # for the same image but with different seasons:
        >>> prompts = ["a landscape with a tree and a river"]*3
        >>> original_images = tools.text2image(prompts)
        >>> edits = ["make it autumn","make it spring","make it winter"]
        >>> all_images, all_prompts = tools.edit_images(original_images, edits)
        >>> score_list, image_list = system.call_classifier(all_images)
        >>> for score, image, prompt in zip(score_list, image_list, all_prompts):
        >>>     tools.display(image, f"Prompt: {prompt}\nConfidence Score: {score}"
            )
        """

class Tools:
    """
    A Python class containing tools to interact with the units implemented in the
        system class,
    in order to run experiments on it.

    Attributes
    ----------
    text2image_model_name : str
        The name of the text-to-image model.
    text2image_model : any
        The loaded text-to-image model.
    images_per_prompt : int
        Number of images to generate per prompt.
    path2save : str
        Path for saving output images.
    threshold : any
        Confidence score threshold for classifier analysis.
    device : torch.device
        The device (CPU/GPU) used for computations.
    experiment_log: str
        A log of all the experiments, including the code and the output from the
            classifier
        analysis.
    exemplars : Dict
        A dictionary containing the exemplar images for each unit.
    exemplars_scores : Dict
        A dictionary containing the confidence scores for each exemplar image.
    exemplars_thresholds : Dict
        A dictionary containing the threshold values for each unit.
    results_list : List
        A list of the results from the classifier analysis.
```

```
    Methods
    -------
    dataset_exemplars(system: System)->List[Tuple[int, str]]
        This experiment provides good coverage of the behavior observed on a
        very large dataset of images and therefore represents the typical
        behavior of the classifier on real images. This function characterizes the
        prototypical behavior of the classifier by computing its confidence score
            on
        all images in the ImageNet dataset and returning the 15 highest confidence
        scores and the images that produced them in Base64 encoded string format.
    edit_images(self, base_images: List[str], editing_prompts: List[str]) -> Tuple[
        List[List[str]], List[str]]
        This function enables localized testing of specific hypotheses about how
        variations on the content of a single image affect classifier confidence
            scores.
        Gets a list of input images in Base64 encoded string format and a list of
        corresponding editing instructions, then edits each provided image based on
            the
        instructions given in the prompt using a text-based image editing model.
            The
        function returns a list of images in Base64 encoded string format and list
            of the
        relevant prompts. This function is very useful for testing the causality of
            the
        classifier in a controlled way, or example by testing how the classifier
            confidence
        score is affected by changing one aspect of the image. IMPORTANT: Do not
            use negative
        terminology such as "remove ...", try to use terminology like "replace ...
            with ..."
        or "change the color of ... to ...".
    text2image(prompt_list: str) -> List[str]
        Gets a list of text prompts as an input and generates an image for each
        prompt using a text to image model. The function returns a
        list of images in Base64 encoded string format.
    summarize_images(self, image_list: List[str]) -> str:
        This function is useful to summarize the mutual visual concept that
        appears in a set of images. It gets a list of images at input and
        describes what is common to all of them.
    describe_images(synthetic_image_list: List[str], synthetic_image_title:List[str
        ]) -> str
        Provides impartial descriptions of images. Do not use this function on
        dataset exemplars. Gets a list of images and generates a textual
        description of the semantic content of each of them.
        The function is blind to the current hypotheses list and
        therefore provides an unbiased description of the visual content.
    display(self, *args: Union[str, Image.Image]):
        This function is your way of displaying experiment data. You must call
        this on results/variables that you wish to view in order to view them.
    """

    def __init__(self, path2save: str, device: str, DatasetExemplars:
        DatasetExemplars = None, images_per_prompt=1, text2image_model_name='sd'):
        """
        Initializes the Tools object.

        Parameters
        ----------
        path2save : str
            Path for saving output images.
        device : str
            The computational device ('cpu' or 'cuda').
        DatasetExemplars : object
            an object from the class DatasetExemplars
```

```python
        images_per_prompt : int
            Number of images to generate per prompt.
        text2image_model_name : str
            The name of the text-to-image model.
        """
    def dataset_exemplars(self, system: System) -> List[Tuple[float, str]]
        """
        This method finds images from the ImageNet dataset that produce the highest
            confidence scores for a specific classifier.
        It returns both the confidence scores and the corresponding exemplar images
            that were used to generate these confidence scores.
        This experiment is performed on real images and will provide a good
            approximation of the classifier behavior.

        Parameters
        ----------
        system : System
            The system representing the specific classifier and layer within the
                neural network.
            The system should have 'layer' and 'classifier_num' attributes, so the
                dataset_exemplars function
            can return the exemplar confidence scores and images for that specific
                classifier.

        Returns
        -------
        List
            For each exemplar image, stores a tuple containing two elements:
            - The first element is the confidence score for the specified
                classifier.
            - The second element is the exemplar images (as Base64 encoded strings)
                 corresponding to the confidence score.

        Example
        -------
        >>> exemplar_data = tools.dataset_exemplars(system)
        >>> for score, image in exemplar_data:
        >>>     tools.display(image, f"Confidence Score: {score}")
        """

    def edit_images(self,
                    base_images: List[str],
                    editing_prompts: List[str]) -> Tuple[List[List[str]], List[str
                        ]]:
        """
        Generates or uses provided base images, then edits each base image with a
        corresponding editing prompt. Accepts either text prompts or Base64
        encoded strings as sources for the base images.

        The function returns a list containing lists of images (original and edited
            ,
        interleaved) in Base64 encoded string format, and a list of the relevant
        prompts (original source string and editing prompt, interleaved).

        Parameters
        ----------
        base_images : List[str]
            A list of images as Base64 encoded strings. These images are to be
            edited by the prompts in editing_prompts.
        editing_prompts : List[str]
            A list of instructions for how to edit the base images derived from
            `base_images`. Must be the same length as `base_images`.
```

```
            Returns
            -------
            Tuple[List[List[str]], List[str]]
                - all_images: A list where elements alternate between:
                    - A list of Base64 strings for the original image(s) from a source.
                    - A list of Base64 strings for the edited image(s) from that source
                        .
                  Example: [[orig1_img1, orig1_img2], [edit1_img1, edit1_img2], [
                        orig2_img1], [edit2_img1], ...]
                - all_prompts: A list where elements alternate between:
                    - The original source string (text prompt or Base64) used.
                    - The editing prompt used.
                  Example: [source1, edit1, source2, edit2, ...]
                The order in `all_images` corresponds to the order in `all_prompts`.

            Raises
            ------
            ValueError
                If the lengths of `base_images` and `editing_prompts` are not equal.

            Examples
            --------
            >>> # test the confidence score of the classifier for the prompt "a
                landscape with a tree and river"
            >>> # for the same image but with different seasons:
            >>> prompts = ["a landscape with a tree and a river"]*3
            >>> original_images = tools.text2image(prompts)
            >>> edits = ["make it autumn","make it spring","make it winter"]
            >>> all_images, all_prompts = tools.edit_images(original_images, edits)
            >>> score_list, image_list = system.call_classifier(all_images)
            >>> for score, image, prompt in zip(score_list, image_list, all_prompts):
            >>>     tools.display(image, f"Prompt: {prompt}\nConfidence Score: {score}"
                )
            >>>
            >>> # test the confidence score of the classifier on the highest scoring
                dataset exemplar
            >>> # under different conditions
            >>> exemplar_data = tools.dataset_exemplars(system)
            >>> highest_scoring_exemplar = exemplar_data[0][1]
            >>> edits = ["make it night","make it daytime","make it snowing"]
            >>> all_images, all_prompts = tools.edit_images([highest_scoring_exemplar]*
                len(edits), edits)
            >>> score_list, image_list = system.call_classifier(all_images)
            >>> for score, image, prompt in zip(score_list, image_list, all_prompts):
            >>>     tools.display(image, f"Prompt: {prompt}\nConfidence Score: {score}"
                )
            """

    def text2image(self, prompt_list: List[str]) -> List[List[str]]:
            """
            Takes a list of text prompts and generates images_per_prompt images for
                each using a
            text to image model. The function returns a list of a list of
                images_per_prompt images
            for each prompt.

            Parameters
            ----------
            prompt_list : List[str]
                A list of text prompts for image generation.

            Returns
            -------
            List[List[str]]
                A list of a list of images_per_prompt images in Base64 encoded string
                    format for
                each input prompts.
```

```
    Examples
    --------
    >>> # Generate images from a list of prompts
    >>> prompt_list = [âĂİJa dog standing on the grassâĂİ,
    >>>                 âĂİJa dog sitting on a couchâĂİ,
    >>>                 âĂİJa dog running through a fieldâĂİ]
    >>> images = tools.text2image(prompt_list)
    >>> score_list, image_list = system.call_classifier(images)
    >>> for score, image in zip(score_list, image_list):
    >>>     tools.display(image, f"Confidence Score: {score}")
    """

def display(self, *args: Union[str, Image.Image]):
    """
    Displays a series of images and/or text in the chat, similar to a Jupyter
        notebook.

    Parameters
    ----------
    *args : Union[str, Image.Image]
        The content to be displayed in the chat. Can be multiple strings or
            Image objects.

    Notes
    -------
    Displays directly to chat interface.

    Example
    -------
    >>> # Display a single image
    >>> prompt = ["a dog standing on the grass"]
    >>> images = tools.text2image(prompt)
    >>> score_list, image_list = system.call_classifier(images)
    >>> for score, image in zip(score_list, image_list):
    >>>     tools.display(image, f"Confidence Score: {score}")
    >>>
    >>> # Display a single image from a list
    >>> prompts = ["a dog standing on the grass"]*5
    >>> images = tools.text2image(prompts)
    >>> score_list, image_list = system.call_classifier(images)
    >>> tools.display(image_list[0], f"Confidence Score: {statistics.mean(
        score_list)}")
    >>>
    >>> # Display a list of images
    >>> prompt_list = [âĂİJa dog standing on the grassâĂİ,
    >>>                 âĂİJa dog sitting on a couchâĂİ,
    >>>                 âĂİJa dog running through a fieldâĂİ]
    >>> images = tools.text2image(prompt_list)
    >>> score_list, image_list = system.call_classifier(images)
    >>> for score, image in zip(score_list, image_list):
    >>>     tools.display(image, f"Confidence Score: {score}")
    """

def summarize_images(self, image_list: List[str]) -> str:
    """
    Gets a list of images and describes what is common to all of them.

    Parameters
    ----------
    image_list : list
        A list of images in Base64 encoded string format.
```

```
        Returns
        -------
        str
            A string with a descriptions of what is common to all the images.

        Example
        -------
        >>> # Summarize a classifier's dataset exemplars
        >>> exemplars = [exemplar for _, exemplar in tools.dataset_exemplars(system
            )] # Get exemplars
        >>> summarization = tools.summarize_images(exemplars)
        >>> tools.display(summarization)
        """

    def describe_images(self, image_list: List[str], image_title:List[str]) -> str:
        """
        Generates textual descriptions for a list of images, focusing
        specifically on highlighted regions. The final descriptions are
        concatenated and returned as a single string, with each description
        associated with the corresponding image title.

        Parameters
        ----------
        image_list : List[str]
            A list of images in Base64 encoded string format.
        image_title : List[str]
            A list of titles for each image in the image_list.

        Returns
        -------
        str
            A concatenated string of descriptions for each image, where each
                description
            is associated with the images title and focuses on the highlighted
                regions
            in the image.

        Example
        -------
        >>> prompt_list = ["a dog standing on the grass",
        >>>                "a dog sitting on a couch",
        >>>                "a dog running through a field"]
        >>> images = tools.text2image(prompt_list)
        >>> score_list, image_list = system.call_classifier(images)
        >>> descriptions = tools.describe_images(image_list, prompt_list)
        >>> tools.display(descriptions)
```

# B  Benchmark models

## B.1  Systems specification

We provide below the full list of objects and categories used for our benchmark.

| | Gender | | Age | |
|---|---|---|---|---|
| | Female | Male | Young | Old |
| 1 | Apron ("kitchen") | Tie | Laptop | Glasses |
| 2 | Umbrella | Beer | Cell phone | Book |
| 3 | Scarf | Skateboard | Skateboard | Hat |
| 4 | Cat | Suit | Bicycle | Tie |
| 5 | Book | Laptop | Teddy bear | Wine glass |
| 6 | Handbag | motorcycle | | |
| 7 | Wine glass | surfboard | | |
| 8 | Hair drier | Frisbee | | |
| 9 | Teddy bear | Guitar | | |
| 10 | Dress | Cap | | |

Table 2: Feature categories and corresponding objects associated with gender and age stereotypes.

| | Color | | | | | Material | |
|---|---|---|---|---|---|---|---|
| | Red | Green | Blue | Black | White | Wooden | Ceramic |
| 1 | Bus | Bus | Bus | Bus | Bus | Table | Vase |
| 2 | Umbrella | Umbrella | Umbrella | Umbrella | Umbrella | Chair | Bowl |
| 3 | Tie | Tie | Tie | Tie | Tie | Bench | Cup |
| 4 | Kite | Kite | Kite | Kite | Kite | | |
| 5 | Frisbee | Frisbee | Frisbee | Frisbee | Frisbee | | |

Table 3: Feature categories and corresponding objects associated with color and material properties.

| | Setting | | | | | | State |
|---|---|---|---|---|---|---|---|
| | Kitchen | Living Room | Office | Wilderness | City | Beach | Misc. |
| 1 | Table | Table | Table | Bird | Bird | Bird | Airplane (Flying) |
| 2 | Chair | Chair | Chair | Car | Car | Car | Bicycle (Ridden) |
| 3 | Cat | Cat | Cat | Dog | Dog | Dog | Clock (Analog) |
| 4 | Dog | Dog | Dog | Horse | Horse | Horse | Keyboard (Typing) |
| 5 | Vase | Vase | Vase | Bench | Bench | Bench | Kite (Flying) |
| 6 | Wine glass | Wine glass | Wine glass | | | | Umbrella (Open) |
| 7 | | | | | | | Vase (With flowers) |

Table 4: Feature categories and corresponding objects associated with different settings and states.

|     | Gender | | Age | |
| --- | --- | --- | --- | --- |
|     | Female | Male | Young | Old |
| 1 | Tie | Apron | Glasses | Laptop |
| 2 | Beer | Umbrella | Book | Cell phone |
| 3 | Skateboard | Scarf | Hat | Skateboard |
| 4 | Suit | Cat | Tie | Bicycle |
| 5 | Laptop | Book | Wine glass | Teddy bear |
| 6 | motorcycle | Handbag | | |
| 7 | surfboard | Wine glass | | |
| 8 | Frisbee | Hair drier | | |
| 9 | Guitar | Teddy bear | | |
| 10 | Cap | Dress | | |

Table 5: Feature categories and corresponding objects with flipped gender and age associations.

## B.2 Dependency strength versus discount factor

We investigate how a *discount factor* $\alpha \in [0, 1]$ of our synthetic model attenuates the score of the synthetic model henever a predefined attribute condition is *not* satisfied. Figure 8 shows the mean classification accuracy for six attribute groups: AGE, COLOR, GENDER, SETTING, SIZE, and STATE.

- **No discount ($\alpha$=1.0).** Baseline accuracies for all groups remain high ($\geq 0.73$).
- **Small discount ($0 < \alpha \leq 0.3$).** Accuracy drops slightly (under five percentage points), indicating that mild penalties leave decisions largely intact.
- **Medium discount ($0.3 < \alpha \leq 0.5$).** Accuracy decreases almost linearlyâĂŤGENDER is most robust, while SIZE and SETTING degrade faster.
- **HighâĂŞextreme discount ($\alpha > 0.5$).** A sharp collapse occurs; COLOR and SIZE fall below 0.20 at $\alpha \approx 0.7$, and all groups eventually saturate between 0.05 and 0.25.

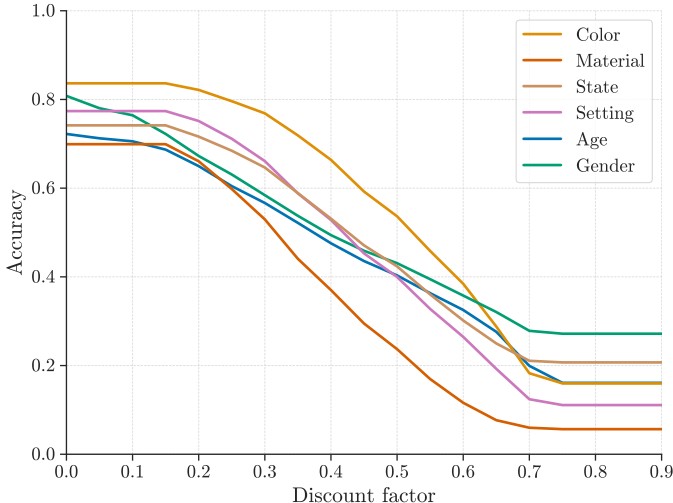

Figure 8: Mean accuracy versus discount factor $\alpha$ for six attribute groups.

# C Results

## C.1 Additional Analysis

Figures 9a and 9b report the average self-evaluation scores across ten rounds of self-reflection, under discount factors of $\alpha$=0.5 and $\alpha$=0.9, respectively. We observe a consistent upward trend in performance across all six attribute categoriesâĂŤGENDER, AGE, COLOR, STATE, and SETTING. This trend holds across both mild and severe reliance scenarios, suggesting that the iterative refinement process is effective in improving the quality of SAIA's hypotheses and explanations. While early rounds exhibit fluctuations (especially at $\alpha$=0.5), later rounds show stabilization and convergence toward higher evaluation scores. The improvement is more pronounced under the lighter discounting condition ($\alpha$=0.5), where SAIA starts from lower scores but achieves a comparable gain. It also noticable that for a disambiguate attribute dependency ($\alpha$=0.5) more self-reflection rounds are necessary, whereas with ($\alpha$=0.9) a saturation is achieved earlier. This demonstrates that self-reflection enables SAIA to recover explanatory accuracy even in challenging scenarios.

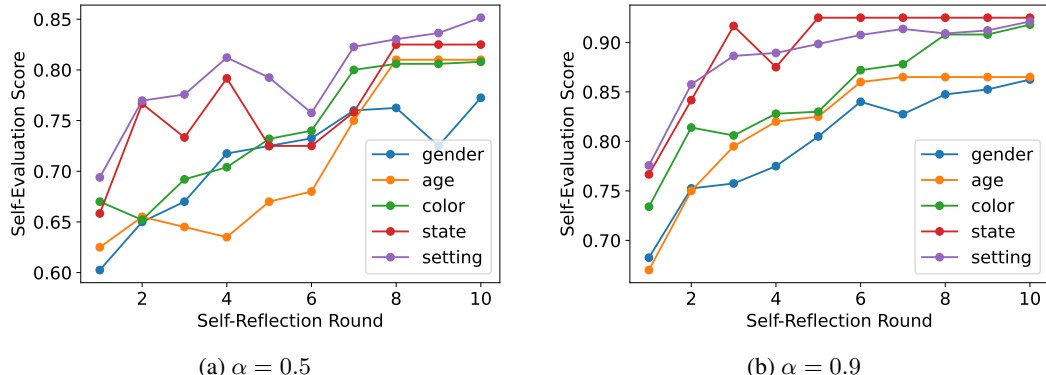

(a) $\alpha = 0.5$           (b) $\alpha = 0.9$

Figure 9: Average self-evaluation scores across ten rounds of self-reflection under different discount factors broken down by reliance attribute category.

## C.2 Multiple feature reliances

For the models that rely on the presence of *both* attributes simultaneously (Table 7), we found that SAIA is be able to accurately uncover both reliances for four out of the five systems, while MAIA was not able to detect both reliances correctly in any of the systems. For the models that rely on the presence of at least one of the two attributes (Table 8, SAIA was only able to recover both reliances for a single model, while MAIA was again not able to detect both reliances for any. While SAIA outperforms MAIA on multi-attribute reliance systems overall, both are less effective that detecting multi-attribute reliances in this second setting. Qualitatively, we notice that for these models, during the hypothesis testing phase, it is more challenging for the agent to isolate the reliance attributes, sometimes identifying other co-occuring attributes instead. For example, for the bench detector that relies on benches that are *wooden* OR in *beach settings*, SAIA concludes that the classifier is biased toward âĂIJtraditional park benches with backrests in natural settings,âĂİ presumably because traditional park benches are often wooden.

## C.3 Text-to-image tool robustness

Text-to-image (T2I) models can carry societal and representational biases. This is a general limitation of agentic interpretability methods that rely on T2I models for generating stimuli. To quantify SAIA's robustness to T2I errors, we conducted two experiments:

**Random failure**: In 50% of T2I calls, we replaced SAIA's prompt with an empty string, resulting in the model generating unrelated content that ignored the intended experimental manipulation. We found that in 80% of these corrupted trials, SAIA successfully noticed the issue, either during the hypothesis-testing stage or self-reflection, and responded by revising its approach or ignoring the incorrect images.

| Target Concept | Attribute 1 | Attribute 2 |
|:---:|:---:|:---:|
| bus | Color: *Red* | Setting: *City* |
| tie | Gender: *Male* | Setting: *Office* |
| bench | Material: *Wooden* | Setting: *Beach* |
| vase | State: *With flowers* | Setting: *Home* |
| apron | Gender: *Female* | Setting: *Kitchen* |

Table 6: Target concept and attribute reliances pairs for the systems with multiple attribute reliances. Each system that relies on the presence of *both* attributes simultaneously will detect the `Target Concept` with high confidence if both `Attribute 1` AND `Attribute 2` are present in the image. Each system that relies on the presence of *at least one* attribute will detect the `Target Concept` with high confidence if either `Attribute 1` OR `Attribute 2` are present in the image.

| | No Reliances Detected | One Reliance Detected | Both Reliances Detected |
|:---:|:---:|:---:|:---:|
| MAIA | 20% | 80% | 0% |
| SAIA | 0% | 20% | 80% |

Table 7: MAIA and SAIA success rates on the multi-attribute reliant systems that rely on the presence of *both* attributes simultaneously.

**Injected bias**: We attack the prompts generated by SAIA in experiments on a tie detector system that relies on the presence of a man. We attack the T2I model by systematically replacing instances of the phrase âĂIJa personâĂİ with âĂIJa manâĂİ $x\%$ of the time for $x = [0, 25, 50, 75, 100]$, and âĂIJa womanâĂİ the remaining $(100 - x)\%$ of the time to simulate a controlled gender bias in the T2I. We found that SAIA was robust to attack and able to detect the correct reliance in all simulated bias ratios. Interestingly, SAIA required more iterations when the T2I model was biased toward generating images of men (i.e. when the bias of the T2I matched the model's feature reliance). We noticed this phenomenon in the base T2I model as well, which almost always generates a man when prompted with âĂIJa person wearing a tie.âĂİ This is another motivating factor for including the study on counterfactual demographic attribute reliant systems (e.g. a tie detector that relies on the presence of a woman)âĂŤsuch systems are out of distribution for not only the multimodal LLM backbone but tools like the T2I as well.

Interestingly, when SAIA notices such unaligned behavior, it usually uses a different experimental design to get the intended behavior (e.g. if the T2I tool doesn't follow the prompt correctly, it uses the editing tool the edit one of the dataset exemplars to achieve the desired stimuli) or ignores the incorrect images and focuses its analysis on the successful generations.

## C.4 Diversity of Hypotheses

We measured the diversity of SAIA's hypotheses by computing the average pairwise similarity between hypotheses generated across all rounds of SAIA for each synthetic benchmark model. Specifically:

- For each system, we parsed all hypotheses produced across rounds.
- We computed the average cosine similarity between the CLIP text embedding of all pairs of hypotheses for that system.
- We then averaged these values across all systems to obtain an overall similarity score.

To contextualize this result, we constructed a baseline using the ground-truth descriptions of all 130 synthetic benchmark models (which vary in both object class and attribute dependence). We computed the average pairwise cosine similarity between these ground-truth descriptions and between SAIA's hypotheses and found that the similarity score for SAIA's hypotheses was 0.073 compared to 0.094 for the baseline. The lower similarity score for SAIA's hypotheses indicates greater diversity than the baseline, suggesting that SAIA explores a wide range of explanations.

|  | No Reliances Detected | One Reliance Detected | Both Reliances Detected |
|---|---|---|---|
| MAIA | 60% | 40% | 0% |
| SAIA | 60% | 20% | 20% |

Table 8: MAIA and SAIA success rates on the multi-attribute reliant systems that rely on the presence of *at least one* attribute.

### C.5    Validation of CLIP and YOLOv8 Reliances on Real Images

We used SAIA's discovered explanations (e.g., reliance on classroom settings to detect the concept of `teacher` or pose to detect the concept of `biker`) to select real images that either contained or lacked the relevant attributes. We then measured the model's responses and found that the same attribute dependencies were reflected in all the real-world image scores, providing strong support for the alignment between synthetic and real domains. The

|  | CLIP Concepts | | | YOLO Concepts | |
|---|---|---|---|---|---|
|  | teacher | scientist | wine glass | biker | pedestrian |
| Generated images | 0.22 / 0.17 | 0.22 / 0.16 | 0.27 / 0.21 | 0.35 / 0.32 | 0.54 / 0.33 |
| Real images | 0.25 / 0.23 | 0.21 / 0.17 | 0.21 / 0.20 | 0.18 / 0.03 | 0.49 / 0.07 |

Table 9: Raw average positive/negative exemplar score pairs for target CLIP and YOLO concepts for which SAIA discovered attribute reliances, evaluated on both generated and real images.

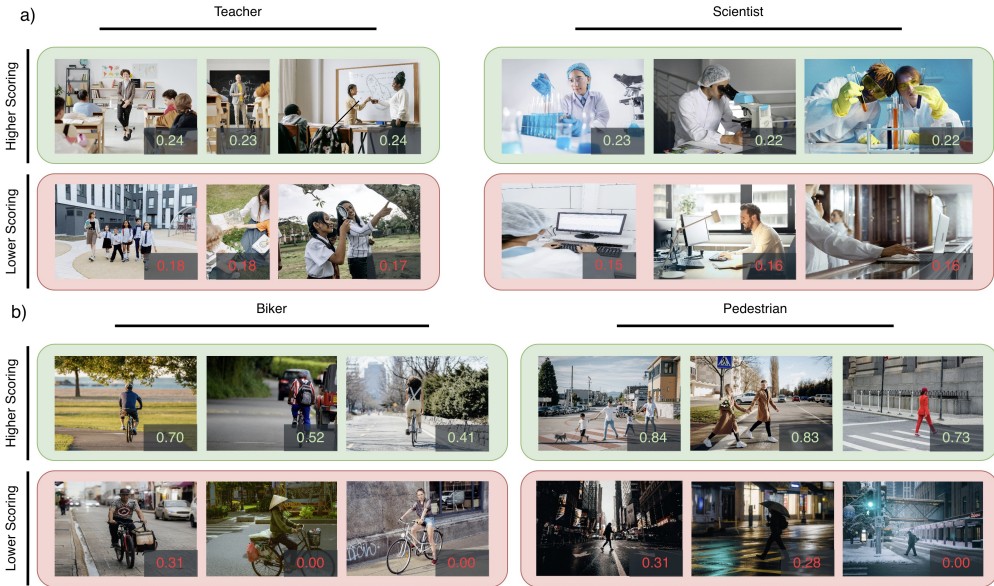

Figure 10: Real images that elicit high (green) and low (red) scores from the (a) CLIP and (b) YOLO classifiers across different object categories. Each triplet shows the top and bottom scoring examples per class for CLIP and YOLOv8.

