# OpenReview forum: "Automated Detection of Visual Attribute Reliance with a Self-Reflective Agent"
_NeurIPS.cc/2025/Conference — NeurIPS 2025 poster_

### Official Review · Reviewer_aRwo · 2025-06-27

**Clarity:** 3
**Significance:** 2
**Originality:** 3
**Rating:** 4
**Confidence:** 3

**Summary:**

Understanding the features utilized by a visual recognition model in its prediction is necessary for detecting improper functionality. While many methods utilized predefined concepts lists and/or manual inspection, the authors propose an automated method leveraging a multimodal LLM. This method incorporates a self-reflective mechanism that separates it from other automated methods and results in improved performance in attribute reliance detection. This is achieved by setting up the LLM to iteratively generate and test hypotheses about a recognition model's reliance on a particular attribute. Additionally, the author curate a benchmark of 130 models that are diversely prepared with reliance on 18 diverse types of visual attributes based on known vulnerability of visual models (e.g., spurious correlations). The models are given predefined reliance thus giving users the ability to assess attribute detection/reliance methods. Finally, the authors demonstrate their automated pipeline's ability to uncover attribute dependencies in pretrained models such as CLIP and YOLO.

**Questions:**

- With the lack of concrete threshold / criterion for when a stable explanation is achieved, will the agent ever determine that the model doesn't rely on unintended attributes, or will it keep trying to find one? Basically, what if the model works as intended?
- The counterfactual setting is unclear to me, could you clarify that intended functionality this experiment is measuring?
- Is having a reliance variable always an issue? For example, when assessing whether an image includes a teacher or scientist, one cannot determine from looking at the person alone, we have to use attributes such as "a class room setting", "a laboratory environment", etc.

**Ethical Concerns:**

["NO or VERY MINOR ethics concerns only"]

**Final Justification:**

Given the many clarifications and expansions to the synthetic models benchmark, and the experiment showing the robustness to failure at the T2I component of the baseline, I'm raising my score to 4.

However, I still think there are limitations to the simplicity of the injected attribute reliance in the synthetic models (based on a single model: Grounding DINO). Additionally, the reliance on the T2I and image editing models may become problematic. Even though you showed that the agent can adapt by "revising its approach or ignoring the incorrect images", it still seems that in these cases the agent may fail to explain the problem with the model (limitation).

**Limitations:**

- Limited by the conceptual understanding and capability by the chosen pretrained multi-modal model
	- The method involves the LLM proposing candidate attributes, generating & editing the images (which would then include accurate understanding of the attributes). If the LLM is capable of accurately classifying, then why use the other model? Are there cases where the LLM fails to classify or detect well, but successfully improves the model it is diagnosing?
- The approach depends on a several larger and more complex models to explain smaller models.
	- Evaluation and and effectiveness is entirely reliant on the strength of the chosen LLMs, summarizers, and image generators.

**Quality:**

2

**Strengths And Weaknesses:**

- (S) Paper organization, problem communication, and method overview are very clear and well written.
- (S) The method successfully detects reliance attributes in the recognition models and outperforms other reliance detection methods.
- (W) The method seems to be reliant on several hyper parameters such as:
	- The choice of multi-modal LLM
	- The choice of text-to-image model
	- The choice of editing method to said text-to-image model
- (S) The addition of a benchmark of 130 models with known attribute reliance is a valuable contribute to reliance detection method, but also explainable & interpretable methods in general
- (W) The attribute reliance of these models is simplistic as it uses an external attribute classifier (either SigLIP or FairFace) to detect the attribute and simply lowers the classification score if the attribute isn't detected. Additionally, all of the 130 models are based on Grounding DINO.
- (W) For the "Counterfactual demographic attributes" described in section 4.2, it is unclear to me what the main difference to "demographic attributes" is.
- (W) The evaluation criteria is entirely dependent on the LLM, so it is unclear when hypothesis testing will end and when self-reflection will end.

---

> ### Author Rebuttal · Authors · 2025-07-31
>
> We thank the reviewer for their meaningful comments! We were glad to see that, like all other reviewers, the reviewer aRwo found our benchmark and synthetic dataset to be useful, and the method's performance to be convincing. Below, we address their comments. To give a short summary of the main points:
> - We clarify the reason for the way we constructed the synthetic benchmarks and its advantages, and provide an initial evaluation for the agent’s ability to recover from an error in the pipeline.
> - We add more complex systems and show the agent is mostly able to work with these. For the final version of the paper we will add an increasingly growing complexity experiment to find the agent’s breaking point.
> - Clarify the definition of the counterfactual systems
> - Clarify the evaluation criteria
> - Add a discussion on whether having a reliance is always an issue—to summarize, we believe it depends on the downstream task.
> - Clarify the reasoning for using large models like LLMs to audit smaller, more specialized models
> ____
>
> **Framework is reliant on several hyperparameters and models**
>
> Indeed, our framework is modular by design, and its behavior may depend on the specific models used as the agent backbone and for the tools. We see this modularity as a strength: as new, more capable tools become available, they can be seamlessly integrated into the framework. To support this, we have:
> Designed the benchmark of 130 synthetic models as a consistent testbed that can be reused to evaluate alternative model configurations
>
> Following the reviewer comment, we conducted the following experiment as an initial evaluation of the framework’s robustness to failures in specific components: in 50% of T2I calls, we replaced the agent’s prompt with an empty string, resulting in the model generating unrelated content that ignored the intended experimental manipulation. We found that in 80% of these corrupted trials, the agent successfully noticed the issue, either during the hypothesis-testing stage or self-reflection, and responded by revising its approach or ignoring the incorrect images. While this is a simple baseline, it gives an approximation of the agent’s sensitivity to breakdowns in the generation pipeline. We are currently expanding this evaluation to simulate degraded versions of each component (e.g., T2I, image editing, image summarization) and will report the agent’s error recovery rate across modules in the appendix of the final version of the paper.
>
> **The reliance of these models is simplistic, as they use an external attribute classifier**
>
> The benchmark models are indeed based on a simplified mechanism. This design was chosen intentionally to allow explicit control over what and how severe ground-truth dependency is, enabling precise and scalable evaluation of whether an interpretability method can recover it. Importantly, this setup allows us to vary the strength of the injected reliance, simulating models with mild to strong biases. In the SM (sec 2.2 and Fig. 1), we show that this mechanism enables us to simulate a wide range of bias levels, from weak reliance to strong or even exclusive reliance. This setup allows us to quantitatively evaluate how well the agent recovers dependencies of varying strength.
>
> A benefit of our mechanism is it enables the construction of more synthetic systems that can be engineered to exhibit a wide range of behaviors. To further extend realism and complexity in the benchmark, we have started to build multi-attribute models with compositional reliance (e.g., concepts that depend on both setting and gender). We observed that the self-reflective agent successfully recovers both dependencies in 80% of systems while MAIA was not able to recover these at all. We will add systems of increasing complexity to test when the agent breaks.
>
> In addition to this controlled benchmark, we validate our method on pre-trained models (e.g., CLIP, YOLOv8), demonstrating that the agent can recover attribute dependencies that emerge organically. These results suggest that our synthetic benchmark complements real-model analyses, providing both a rigorous evaluation ground and a path for future expansion.
>
> **What are the “Counterfactual demographic attributes”?**
>
> Demographic attribute systems simulate reliance on naturally co-occurring demographic signals (e.g., the presence of a male face often coinciding with ties in training data). These are designed to reflect the types of biases commonly found in real-world models and datasets. Counterfactual demographic systems, by contrast, are constructed to simulate atypical or out-of-distribution dependencies e.g. a model that detects a tie only in the presence of a person who is female, which rarely co-occurs in real-world data. These systems allow us to test whether the agent can detect unexpected or counterintuitive reliance patterns that do not follow natural co-occurrence statistics. This distinction allows us to assess both the agent’s ability to uncover realistic demographic biases and its robustness to rare or previously unknown dependencies. We will make this difference more explicit in the final version of the paper.
>
> **The evaluation criteria are entirely dependent on the LLM**
>
> While it is true that the agent is built from a multimodal LLM to reason over experimental outcomes, its decision process is guided by a structured protocol that provides consistency.
> In more detail: In the hypothesis-testing phase, the agent runs a number of experimental rounds, where each round involves proposing a hypothesis, designing targeted stimuli (via prompting or editing), and evaluating the model’s behavior. The self-reflection stage follows a fixed diagnostic protocol: the winning hypothesis from the previous hypothesis-testing stage is passed to a separate LLM that generates two diverse sets of prompts predicted to elicit high and low detection scores. Then we measure the model scores on these images. If the resulting score distributions do not align with the agent's expectations (i.e., little to no score separation between the two sets of images), the agent concludes that the hypothesis lacks support and re-enters the hypothesis-testing phase with a revised prompt.
>
> We incorporate two concrete mechanisms to guide and constrain the agent's decision process:
>
> - Empirical Threshold for Self-Evaluation: In the self-reflection stage, the agent compares model scores between two groups of synthetic images: those predicted to yield high scores and those predicted to yield low scores under the current conclusion. If the mean scores between the two groups are not sufficiently separated based on an empirically set threshold (i.e. the conclusion is not sufficiently discriminative), the agent concludes that the hypothesis lacks support and initiates a new hypothesis-testing round.
>
> - Maximum Round Limit: We cap the total number of agent rounds (hypothesis-testing + self-reflection) to 10. If no hypothesis passes the self-evaluation threshold by that point, the agent returns the hypothesis that achieved the best self-evaluation score (typically the most recent one). In practice, the agent usually converges before reaching this cap. This behavior is visualized in Figure 5, which shows that the self-evaluation score tends to improve monotonically across rounds.
>
> We will make sure to clearly explain this in the method section.
>
> **Will the agent ever determine that the model doesn't rely on unintended attributes**
>
> Great point! Yes, in some cases the agent concludes that there are no dependencies, and we will include such examples in the supplements of the paper.
>
>
> **Is having a reliance variable always an issue?**
>
> Our framework does not attempt to judge whether a discovered reliance is inherently harmful or contextually appropriate, nor is this distinction communicated to the agent. Rather, the goal is to surface potential dependencies that might not have been previously anticipated, enabling practitioners to make informed judgments based on their specific downstream use case. In some applications, reliance on context (e.g., background for "teacher") may be a valid, even desirable, part of the concept's representation. In others (e.g., medical imaging or fairness-sensitive applications), such dependencies could lead to unwanted generalization or bias.
>
> **Why not use the backbone LLM for the recognition task itself?**
>
> The large models used in our framework are too large and expensive to be used in some practical deployment scenarios, such as on-device inference or embedded systems. Many real-world models are much smaller and more efficient. Our approach allows to use the strengths of foundation models, like strong semantic understanding and language grounding, to systematically identify dependencies in other models that might be less capable, more compact, and even task-specific.

---

> > ### Author Response · Authors · 2025-08-05
> >
> > We would like to thank the reviewer again for their feedback. As the discussion period is coming to a close, we wanted to check whether our response addresses your concerns. We're happy to provide further clarification if needed.

---

> > ### Comment · Reviewer_aRwo · 2025-08-05
> >
> > ### Framework is reliant on several hyperparameters and models
> > - I agree the modularity is a strength of this model, and the contribution of the 130 synthetic models will provide a case of comparison to other approaches.
> > - Thank you for the additional experiment showing the agent's ability to handle failure cases, specifical with the T2I model.
> >
> > ### The reliance of these models is simplistic, as they use an external attribute classifier
> > - You have shown that the agent is capable on non-simplistic areas, but your benchmark is still simplistic in nature
> > - Thank you for expanding the complexity of the synthetic models with compositional reliance, however, the injection of reliance at a layer is different than deeply embedded reliance within the whole network
> > - With that said, this is still a solid contribution with some limitations
> >
> > ### What are the “Counterfactual demographic attributes”?
> > - Thank you for this clarification!!
> >
> > ### The evaluation criteria are entirely dependent on the LLM & Will the agent ever determine that the model doesn't rely on unintended attributes & Is having a reliance variable always an issue?
> > - Thank you for clarifying these!
> >
> > ### Why not use the backbone LLM for the recognition task itself?
> > - This makes sense as an offline evaluation framework for online deployed models as you've argued. Thank you!
> >
> >
> > Given the many clarifications and expansions to the synthetic models benchmark, and the experiment showing the robustness to failure at the T2I component of the baseline, I'm raising my score to 4.
> >
> > However, I still think there are limitations to the simplicity of the injected attribute reliance in the synthetic models (based on a single model: Grounding DINO). Additionally, the reliance on the T2I and image editing models may become problematic. Even though you showed that the agent can adapt by "revising its approach or ignoring the incorrect images", it still seems that in these cases the agent may fail to explain the problem with the model (limitation).

---

### Official Review · Reviewer_H1cw · 2025-07-02

**Clarity:** 3
**Significance:** 3
**Originality:** 3
**Rating:** 5
**Confidence:** 3

**Summary:**

This paper proposes an automatic framework for detecting which visual attributes an object recognition model uses when identifying a visual concept, especially focusing on detecting unintended attributes. The framework uses a "self-reflective agent" that works in two stages.

First, in the "Hypothesis-Testing" stage, the agent (based on Claude-Sonnet-3.5) is provided with a subject recognition model and a target visual concept and must discover on which visual attributes on the input image the model relies on in order to recognize the target visual concept. To that end, the agent proposes test hypotheses about the attributes, generates synthetic images with and without these suspected attributes, scores the images with the subject recognition model, and refines its proposed hypotheses based on the model's responses until it reaches a "stable explanation".

Next, in the "Self-Reflection" stage, another language model evaluates the explanation from the first stage. It checks if the explanation matches the recognition model’s confidence scores by again generating and scoring (with the model) more synthetic images. If inconsistencies are found, it launches a new "Hypothesis-Testing" stage.

The paper introduces a new benchmark of synthetic models with controlled attribute dependencies to evaluate the framework. Experiments show that the framework improves with more self-reflection stages. It is also tested on real models like CLIP and YOLOv8, successfully identifying their visual attribute dependencies.

**Questions:**

Please answer my concerns/questions/concerns listed on the weaknesses section above. Specifically:
- How do you respond on my comment about the assumption that the recognition model depends on a single, "decisive" visual attribute?
- How well the detected visual attributes from the synthetic images match the model's behavior on real-world images
- What is the difference between the testing phase in the "Hypothesis-Testing" stage and the "Self-Reflection" stage
- Questions about workings of the proposed framework

**Ethical Concerns:**

["NO or VERY MINOR ethics concerns only"]

**Final Justification:**

My initial review of this work was positive, highlighting its novelty, well-designed framework, and the introduction of a novel benchmark. However, I raised several concerns: 1) the assumption that a subject recognition model relies on a single decisive visual attribute, 2) the unclear behavior of the model on real-world images (i.e., whether the generated explanations align with subject model outcomes on the these real-world images), 3) confusion about the difference between the "Hypothesis-Testing" and "Self-Reflection" stages, and 4) the framework’s apparent complexity.

The rebuttal addressed most of these concerns effectively. The authors described additional experiments that demonstrate that their framework’s explanations match subject model behavior on real-world images and clarified the distinction between the two stages. However, further experiments during the discussion period revealed a limitation: the framework struggles with subject models that exhibit high confidence when relying on multiple attributes.

Despite this limitation, I still find the work interesting and valuable. The authors’ commitment to additional evaluations will help clarify the operational scope of their system and may inspire future research to address this issue. Based on the authors’ responses, it also appears that they have addressed the other reviewers’ concerns to a reasonable extent.

Thus, I maintain my initial positive assessment and recommend acceptance.

**Limitations:**

Yes

**Quality:**

3

**Strengths And Weaknesses:**

**Strengths:**
- The paper introduces a novel agentic framework for automated interpretation of vision models (i.e., automatically detecting the visual attributes on which they rely on), addressing an important and emerging research direction in explainable AI.
- The proposed framework appears to be well-designed and makes (overall) sense. The results demonstrate the advantage of integrating into the framework "self-reflection" stages.
- Also, it is supported by a novel benchmark, which can be proved useful for future works in this area.
- The paper is generally well-written and accessible, successfully communicating its key ideas (however, as discussed below, some descriptions lack technical clarity)

**Weaknesses:**
- The work appears to assume that the recognition model depends on a single, "decisive" visual attribute (meaning the model succeeds only if the attribute is present and fails otherwise). However, in practice, recognition models might rely on multiple attributes, and their decisions may depend on a combination of features rather than a single absolute factor. For instance, a model might recognize a concept when a sufficient subset of relevant attributes is present, rather than requiring all or none.
- Based on the above, and given that the framework’s reliance on synthetic images, it remains unclear how well the generated explanations align with the model’s behavior on real-world images (if I understood correctly, both the proposed framework and the proposed benchmark rely on synthetic images for providing explanations and evaluating them respectively). Did the authors test whether the generated explanations (e.g., for the real models CLIP and Yolov8) match the model outcomes on real-world images instead of synthetic ones?
- It is not clear what is the difference between the testing phase in the "Hypothesis-Testing" stage and the "Self-Reflection" stage. Both generate synthetic images with and without the visual attribute, then score these images with the recognition model, and observe if the hypothesis matches the model's score outcomes. So, why a different "Self-Reflection" stage is needed is unclear.
- While on high-level it is easy to understand how the approach works, the main paper's description of how exactly the two main stages work is not entirely clear. For instance, it is not clear how exactly the agent determines whether a hypothesis matches the observed results in the "Hypothesis-Testing" and "Self-Reflection" stage. What is the exact criterion for determining if the hypothesis and the results match or requires further refinement?  Does the framework relies exclusive on synthetic images for generating explanations about the visual attributes?
- The proposed framework appears to be somewhat complex.

---

> ### Author Rebuttal · Authors · 2025-07-31
>
> We thank the reviewer for their meaningful comments! We were glad to see that, like all other reviewers, the reviewer H1cw found our benchmark and synthetic dataset to be useful and the self-reflection to be a meaningful contribution. Below, we address their comments. To give a short summary of the main points:
> - We constructed additional systems with dual-attribute dependency and showed that the agent can successfully identify both attributes. We will add these and more complex systems with multiple attribute reliances to the final version of the paper.
> - We evaluated the findings from the agent on real images and found the behavior to be as expected. We will add this evaluation to the appendix of the paper.
> - We clarify the different between the different stages and the stopping criteria
> ____
>
> **Assumption for a single, "decisive" visual attribute**
>
> Following this comment we created 5 more synthetic benchmark models that have 2 attribute reliance (e.g. a bench detector that relies on *wooden benches* in *beach* settings) and found the self-reflective agent to be able to accurately uncover both reliances for 4 out of the 5 systems, while MAIA was not able to detect both reliances correctly in any of the systems. We will create more systems with multiple attribute reliances (more than 2) and report the tradeoff between performance and reliance-complexity in the appendix of the paper.
>
> We would like to note that even in the benchmark systems that have only one attribute reliance, the agent commonly initially concludes reliance on multiple attributes and narrows down its conclusion over further rounds of self-reflection and hypothesis-testing. For example, in the provided supplemental experiment folder “black_umbrella_0.9”, we provide the full trace of an experiment on a *umbrella* detection model with an injected bias towards black umbrellas. After the first hypothesis-testing round (“round0.html”), the agent concludes the following: *“The classifier has a strong bias towards black umbrellas being used for rain protection in moody urban settings.”* In the subsequent self-reflection and hypothesis-testing rounds, this conclusion is found to be overspecified, and the agent ultimately discovers the more general ground truth reliance: *“The classifier identifies umbrellas primarily through their complete geometric form and prototypical appearance (especially black color)”*
>
> **How well the generated explanations align with the model’s behavior on real-world images?**
>
> Great point! Following the reviewer’s suggestion, we conducted additional evaluations using real images for all attributes found for CLIP and YOLOv8. In these evaluations, we used the agent’s discovered explanations (e.g., reliance on classroom settings or biker pose) to select real images that either contained or lacked the relevant attributes. We then measured the model’s responses and found that the same attribute dependencies were reflected in all the real-world image scores, providing strong support for the alignment between synthetic and real domains. We will include these new results in the final version of the paper and are currently working on a larger-scale validation using real-world image datasets to systematically assess the generalizability of explanations.
>
> **The difference between the testing phase in the "Hypothesis-Testing" stage and the "Self-Reflection" stage.**
>
> Thank you for raising this important question. While both stages involve generating and evaluating synthetic images, their roles, timing, and structure within the agent’s reasoning process are fundamentally different.
>
> - Hypothesis-Testing Stage: This stage is part of an open-ended, exploratory process. The agent iteratively proposes hypotheses and actively designs multiple experiments to test them, this may involve generating contrasting image pairs, editing dataset exemplars, or manipulating visual attributes in various ways. The agent then analyzes the pattern of scores across these experiments to decide whether the hypothesis is supported. Crucially, the agent may test several hypotheses during this stage.
>
> - Self-Reflection Stage: This stage occurs after the agent has reached a conclusion. The agent now uses a standardized self-evaluation protocol: it calls another LLM that takes the conclusion regarding the attribute reliance and generates 10 image prompts predicted to elicit high scores (with the attribute) and 10 image prompts that should elicit low scores (without it). These prompts are then fed to a T2I model, and the generated images are fed to the subject model to measure whether the model’s responses match this predicted contrast. This protocol acts as an external consistency check, validating whether the conclusion generalizes. If it fails, the agent reflects on the mismatch and re-enters the hypothesis-testing loop with updated insights.
> This structure is essential to enable the agent to self-correct: if the self-reflection stage reveals inconsistencies between the conclusion and model behavior, the agent revises or rejects the conclusion entirely. We will clarify this distinction more clearly in the final version of the paper.
>
> **How does the agent determine whether a hypothesis matches the observed results**
>
> We incorporate two concrete mechanisms to guide and constrain the agent's decision process:
>
> - Empirical Threshold for Self-Evaluation: In the self-reflection stage, the agent compares model scores between two groups of synthetic images: those predicted to yield high scores and those predicted to yield low scores under the current conclusion. If the mean scores between the two groups are not sufficiently separated based on an empirically set threshold (i.e. the conclusion is not sufficiently discriminative), the agent concludes that the hypothesis lacks support and initiates a new hypothesis-testing round.
>
> - Maximum Round Limit: We cap the total number of agent rounds (hypothesis-testing + self-reflection) to 10. If no hypothesis passes the self-evaluation threshold by that point, the agent returns the hypothesis that achieved the best self-evaluation score (typically the most recent one). In practice, the agent converges before reaching this cap. This behavior is visualized in Figure 5, which shows that the self-evaluation score tends to improve monotonically across rounds.
>
> We will make sure to clearly explain this in the method section.
>
>
> **Does the framework rely exclusively on synthetic images?**
>
> The method relies on both looking at activation of images from a fixed dataset of real images (“dataset exemplars”) to get an initialization for the search, and then on generated synthetic images that test different aspects of the hypotheses. In order to make sure that the conclusions are valid, we conducted additional evaluations using real images for all attributes found for CLIP and YOLOv8 and found that the same attribute dependencies were reflected in all the real-world images.

---

> > ### Comment · Reviewer_H1cw · 2025-08-04
> > **Response to rebuttal**
> >
> > Thank you for answering my comments. The responses resolve some of my concerns, though one key issue remains only partially addressed.
> >
> > My original concern was about the recognition model relying on (1) a single attribute and (2) a "decisive" attribute (where the model outputs high confidence score when the attribute is present and low otherwise). The authors tested their approach with two attributes, reporting success in 4 out of 5 cases—which partially addresses the first point (however, I have a follow-up question: In these successful cases, did the model’s reliance conclusion explicitly list both attributes?).
> >
> > The second part of my concern—about cases where multiple attributes contribute (rather than a single decisive one)—was not fully addressed. For example, if there are three possible attributes, and the model outputs high confidence when at least one is detected, how does the approach handle this?

---

> > > ### Author Response · Authors · 2025-08-05
> > > **Thank you for the response**
> > >
> > > We thank the reviewer for their reply and the clarifications.
> > >
> > > In our first response, we constructed 5 new synthetic benchmark models that rely on the presence of two attributes simultaneously (e.g. a bench detector that relies on benches that are wooden AND in beach settings) and found the self-reflective agent to be able to accurately uncover both reliances for 4 out of the 5 systems. And the 4 successful cases, the self-reflective agent’s conclusion explicitly lists the reliance on **both features**, while a non self-reflective agent like MAIA is only able to uncover one or none of the reliances. For the convenience of the reviewer, we include below the full results for all 5 systems with their ground-truth descriptions, self-reflective agent results, and MAIA results.
> > >
> > > The reviewer brings up an important question about the agent’s behavior for systems in which the model has high confidence as long as *at least one* of multiple reliances are present (e.g. a bench detector that relies on benches that are wooden OR in beach settings). As an initial evaluation of this setting, we modified our 5 synthetic models with two attribute reliances from the experiment described above to output high confidence if at least one of the reliances is present. This initial evaluation reveals that this is indeed a more challenging setting for the agent; it was only able to recover both reliances for a single model. In 1 of the other models, it recovers 1 of the reliances, and in the remaining 3 models, it recovers none. Qualitatively, we notice that for these models, during the hypothesis testing phase, it is more challenging for the agent to isolate the reliance attributes, sometimes identifying other co-occuring attributes instead. For example, for the bench detector that relies on benches that are wooden OR in beach settings, the agent concludes that the classifier is biased toward “traditional park benches with backrests in natural settings,” presumably because traditional park benches are often wooden. We agree that this is an interesting case, and we will add a full evaluation of it to the paper.

---

> > > > ### Author Response · Authors · 2025-08-05
> > > > **Results - Synthetic benchmark models with two reliances ("AND" case)**
> > > >
> > > > **Model 1**
> > > >
> > > > *Ground truth:* The *bus* classifier relies on buses that are red AND in city settings.
> > > >
> > > >
> > > > *MAIA response:* The classifier should identify buses generally, but instead exhibits a strong bias toward red-colored buses, while significantly underscoring buses of other colors regardless of their design authenticity.
> > > >
> > > > *Self-reflective agent response:* The classifier should identify buses generally, but instead exhibits a strong bias toward red London-style buses in urban street contexts, requiring both the complete bus form and proper street infrastructure elements for highest confidence scores.
> > > >
> > > > **Model 2**
> > > >
> > > > *Ground truth:* The *tie* classifier relies on ties being worn by males AND in office settings.
> > > >
> > > > *MAIA response:* The classifier exhibits a strong bias toward solid-colored ties in professional business contexts with proper upper-body framing, while being surprisingly tolerant of variations in attire formality and face visibility.
> > > >
> > > > *Self-reflective agent response:* The classifier exhibits a strong bias toward traditionally masculine, middle-aged professionals in formal business attire photographed frontally in office settings, while systematically discounting ties worn in any other context, by any other demographic, or photographed from any other angle.
> > > >
> > > > **Model 3**
> > > >
> > > > *Ground truth:* The *bench* classifier relies on benches that are wooden AND in beach settings.
> > > >
> > > > *MAIA response:* The classifier has a strong bias toward traditional wooden park benches with visible slat patterns, showing reduced performance on modern, ornate, or occupied benches despite their valid bench functionality.
> > > > *Self-reflective agent response:* The classifier demonstrates a specific bias toward “vista point benches” – elevated wooden benches positioned for contemplative ocean viewing with clear horizon visibility and natural landscape integration.
> > > >
> > > > **Model 4**
> > > >
> > > > *Ground truth:* The *vase* classifier relies on flowers present in the vase AND in home settings.
> > > >
> > > > *MAIA response:* The classifier exhibits a narrow bias toward vases containing structured flowers photographed on windowsills with natural lighting, rather than recognizing vases in diverse contexts and arrangements.
> > > >
> > > > *Self-reflective agent response:* The classifier should identify vases, but shows a strong bias towards dense flower arrangements photographed in well-lit indoor settings with visible windows and furniture context.
> > > >
> > > > **Model 5 (Unsuccessful Response)**
> > > >
> > > > *Ground truth:* The *apron* classifier relies on aprons being worn by females AND in kitchen settings.
> > > >
> > > > *MAIA response:* The classifier exhibits a strong bias towards active cooking scenarios, particularly food preparation activities, rather than simply detecting the presence of aprons.
> > > >
> > > > *Self-reflective agent response:* The classifier should detect aprons, but it has a strong bias towards women in casual clothing wearing light-colored aprons while actively preparing food, regardless of kitchen setting type.

---

> > > > ### Comment · Reviewer_H1cw · 2025-08-07
> > > > **Response to authors**
> > > >
> > > > Thank you again for addressing my follow-up comments. While the current approach seems to struggle with subject models that have high confidence when at least one of multiple reliances are present, I still find the work interesting and valuable. The additional evaluations on this issue (promised by the authors) will help clarify the operational scope of the proposed system and may inspire future research to address it.

---

> > > > > ### Author Response · Authors · 2025-08-07
> > > > >
> > > > > We thank the reviewer for their meaningful and encouraging feedback. We will ensure that the additional evaluation and discussion are included in the next version of the paper.

---

### Official Review · Reviewer_Cd4M · 2025-07-02

**Clarity:** 3
**Significance:** 2
**Originality:** 2
**Rating:** 4
**Confidence:** 3

**Summary:**

This paper proposes an automated agent for visual attribute reliance. The idea is to use an LLM agent that can propose hypothesis, tests them and iteratively refines them based on observations. One key contribution of the method is the self-reflection stage in which the agent uses a self-evaluation tool to score it's explanation. The authors also construct a synthetic dataset to measure the effectiveness of their proposed approach.

**Questions:**

1. Is there any error analysis that is done to show how sensitive or robust the method is when text to image models fail to generate the presence or absence of some attributes. How much error can we tolerate.
2. T2I models can suffer when generating some rare classes, like rare breeds of dog etc. How can the self-reflective agent work in these settings.
3. Can the LLMs come up with diverse set of candidate hypothesis? Is there a way to quantify this?

**Ethical Concerns:**

["NO or VERY MINOR ethics concerns only"]

**Final Justification:**

The reviewer clarified my concern on over-reliance of T2I models. As mentioned by the reviewer, the self-reflective agent can help mitigate such biases due to an external LLM judge. Since my misunderstanding is clarified from the rebuttal, I increase the score to borderline accept.

**Limitations:**

I feel the contribution over MAIA is limited. The over-reliance on text to image generators is a bit concerning when they themselves can have biases or failure modes.

**Quality:**

2

**Strengths And Weaknesses:**

Strengths:
1. The use of self-reflection agent is quite interesting. It is a natural extension to MAIA style approaches.
2. The proposed synthetic dataset can be useful. The method for constructing the dataset makes sense.

Weakness:
1. Main limitation is that the contribution is limited. I feel the use of agents for visual explanations have been proposed in works like MAIA. This work just adds a self reflecting agent on top of it.
2. The self-reflecting agent assumes text to image model is good. But text to image model can have biases in itself. They might not be able to generate the presence or negation of some attributes. T2I models have especially been shown to be poor at negations. Given this, it is hard to conclusively tell how good the self-reflection stage is.
3. There are a lot of components cascaded in this system. So, if one stage breaks, the entire system can fail. This concerns me.
4. The diversity of generations from T2I models can also be low. Is that a concern?
5. The synthetic dataset is constructed with simple object classes and attributes. What about the long tail category objects?

---

> ### Author Rebuttal · Authors · 2025-07-31
>
> We thank the reviewer for their meaningful comments! We were glad to see that, like all other reviewers, the reviewer Cd4M found our benchmark and synthetic dataset to be useful, and the self-reflection to be a meaningful contribution. Below, we address their comments. To give a short summary of the main points:
>
> - We clarify the contribution over MAIA
> - We clarify the usage of T2I models in the self-evaluation phase. Instead of prompting a T2I with an attribute and its negation, we use a separate LLM to generate diverse prompts that we find to be much more robust to inaccurate generations from the T2I tool.
> - We provide an initial evaluation of the ability of the agent to recover from failures of tools
> - We conducted additional experiments on ResNet-152 for classifying rare classes in ImageNet
> - We provide a quantitative evaluation of the diversity of the hypotheses
> ___
> **Limited contribution compared to MAIA**
>
> Our work builds upon MAIA by introducing a *self-reflective loop*, which we believe is a significant step forward, both in terms of methodological design and empirical performance.
>
> While MAIA concludes the experiments after a single hypothesis-testing pass, our agent critically evaluates the accuracy of its own conclusions and initiates further experimentation when inconsistencies arise. This enables the agent to revise or refine hypotheses, correct misattributions, and uncover more robust explanations. For example, as shown in Fig. 2, the agent initially concludes that a model relies on *“corporate settings”* to detect suits, but revises this to *“male-presenting figures wearing suits”* after it measures behavior in the self-evaluation phase didn’t follow the predicted one. The reviewer can find many more results like this in the provided html files in the supplementary material.
> Quantitatively, this reflective capability leads to substantial gains: across our benchmark (Fig. 5a), self-reflection improves predictiveness scores significantly beyond MAIA, and even outperforms ensembles of MAIA-like agents. This suggests that the improvement yields systematic advantages in model auditing.
>
> **T2I tools might be biased**
>
> We agree that T2I models can carry societal and representational biases. This is indeed a general limitation of agentic interpretability methods that rely on T2I models for generating stimuli (and was noted and evaluated in the MAIA paper, see their section 4.4).  However, a key distinction in our framework is the inclusion of a self-reflection loop, which we find makes the system more robust to such artifacts.
>
> For example, in the provided supplemental experiment folder “male_cap_0.5”, we provide the full trace of an experiment on a *cap* detection model with an injected bias towards men wearing caps. We observe that the T2I model exhibits gender bias: despite gender-neutral prompts (e.g. “a person wearing a black baseball cap”), all generated images in the first hypothesis-testing round (“round0.html”) depict men wearing caps. As a result, the agent is initially unable to draw any conclusions about potential gender dependencies through hypothesis-testing alone. During self-reflection in round3.html, the agent is evaluating the hypothesis: *“The classifier demonstrates a strong contextual bias toward caps being actively worn on human heads…”* It does so by passing the conclusion to a separate LLM that generates two diverse sets of prompts predicted to elicit high (e.g. *“A young man wearing a red baseball cap while smiling at the camera”*, *“Woman jogger wearing a white sports cap during her morning run”*, *“Teen boy wearing his baseball cap backwards at a skate park”*) and low (e.g. *“New baseball cap displayed on store shelf”*, *“Collection of caps hanging on wall hooks”*, *“Baseball cap floating in swimming pool”*) detection scores, respectively. As a result of the diverse evaluation prompts, the agent is able to recognize a discrepancy in its previous conclusion and formulates a new hypothesis: *“The classifier exhibits a gender and age bias, favoring adult males wearing baseball caps in athletic or casual contexts.”* This insight leads the agent to revise its prompts and explicitly query for images of women wearing caps, ultimately uncovering the correct dependency, despite the T2I model's initial bias.
>
> To quantify the agent’s robustness to T2I biases, we conducted an experiment to deliberately inject a bias to the T2I model and show that the agent was able to recover the correct reliance despite it across different levels of injected biases. We invite the reviewer to refer to the reply to reviewer wjA6 above **"T2I models have their own biases"** for the experimental setting and results.
>
> **The T2I model might not be able to generate the presence or negation of some attributes**
>
> To clarify, the self-evaluation protocol does *not* operate by prompting the T2I model with a minimal pair of prompts (e.g. *“a cap on a human head”* vs. *“a cap not on a human head”*). Rather, as described in the previous section, the agent passes its current conclusion to a separate LLM, which then generates two diverse sets of prompts predicted to elicit high and low detection scores, respectively. We thank the reviewer for highlighting this unclear description of the self-evaluation protocol and will incorporate a more comprehensive explanation in the revision.
>
> We found this strategy to be more robust compared to naive prompting with direct negations (which we indeed found prone to semantic and visual drift). We refer the reviewer to the reply to reviewer wjA6 above **"Image generation in the self-evaluation might be inaccurate"** for more discussion about the effect of T2I failure on self-reflection.
>
> **Diversity of T2I**
>
> Please note that we never generate multiple samples from the same prompt. Instead, in the self-reflection stage we ask a language model to generate diverse sets of prompts corresponding to the presence or absence of the detected attribute (as described in Section 3.2). Each prompt is sampled once, so the resulting image set reflects conceptual diversity across different instantiations of the attribute, rather than relying on stochastic sampling from a fixed prompt. We invite the reviewer to explore the submitted html files to judge the diversity of generated images.
>
> **Rare Classes**
>
> When investigating attribute dependencies in ResNet-152 for classifying rare ImageNet classes, we found that the agent was able to effectively reason about certain classes, e.g. loupe, and tools like the T2I generated reliable outputs. However, for other classes, such as the pickelhaube, the T2I was only able to reliably generate pickelhaubes in isolation. When prompted to generate a pickelhaube worn by a soldier, the T2I would generate a regular army helmet, leading to low scores from the model and the agent to conclude that the model has a bias against pickelhaubes worn by humans. Thus, in rare classes, it is true that it may be difficult for the agent to disentangle model attribute dependencies from nuanced failures of the T2I—this is a limitation of the framework that we will discuss further in the final version of the paper. In cases where the subject model operates on uncommon concepts, the users might adjust the tools for the agent.
>
> **Components cascade can cause the framework to break:**
>
> We agree that cascading multiple components might introduce the risk of compounding errors, which is a known challenge in agentic frameworks. However, our self-reflective agent is better at addressing this concern. The self-evaluation protocol systematically compares predicted and actual model behavior, allowing the agent to detect inconsistencies that may result from failures in earlier stages. When such discrepancies are found (whether because of a misinterpretation of the agent or a failure of one of the tools), the agent re-enters the hypothesis-testing loop and attempts to revise or refine its explanation. This feedback loop improves the system’s robustness by preventing premature or faulty conclusions from being returned to the user.
>
> To quantify the agent’s robustness to failure of one of its tool, we conducted the following experiment: We intentionally elicit failures in the T2I tool such that in 50% of T2I calls, we replaced the agent’s prompt with an empty string, resulting in the model generating unrelated content that ignored the intended experimental manipulation. We found that in 80% of these corrupted trials, the agent successfully noticed the issue, either during the hypothesis-testing stage or self-reflection, and responded by revising its approach or ignoring the incorrect images. We plan to expand this evaluation by introducing more targeted “tools attacks” and we will include the results in the appendix of the paper.
>
> **Are the hypotheses diverse? Can we quantify this?**
>
> We conducted a quantitative analysis to measure hypothesis diversity: we computed the average pairwise cosine similarity between hypotheses generated across all rounds of the self-reflective agent for each synthetic benchmark model seperatly. Specifically:
>
> - For each system, we parsed all hypotheses produced across rounds.
> - We computed the average cosine similarity between all pairs of hypotheses for that system.
> - We then averaged these values across all systems to obtain an overall similarity score.
>
> We also constructed a baseline using the ground-truth descriptions of all 130 synthetic benchmark models (which vary in both object class and attribute dependence). We computed the average pairwise cosine similarity between these ground-truth descriptions.
>
> *Results:*
> - Agent hypotheses similarity score: 0.073
> - Ground-truth similarity score: 0.094
>
> The lower similarity score for the agent’s hypotheses indicates greater diversity than the baseline, suggesting that the agent explores a wide range of explanations. We invite the reviewer to explore the agent's hypotheses in the experiments provided in the SM.

---

> > ### Author Response · Authors · 2025-08-05
> >
> > We would like to thank the reviewer again for their feedback. As the discussion period is coming to a close, we wanted to check whether our response addresses your concerns. We're happy to provide further clarification if needed.

---

### Official Review · Reviewer_wjA6 · 2025-07-03

**Clarity:** 3
**Significance:** 2
**Originality:** 2
**Rating:** 4
**Confidence:** 4

**Summary:**

The authors introduce an agentic framework for identifying visual attribute dependencies in trained vision models. The core of the contribution is a "self-reflective agent," implemented with a multimodal LLM, that systematically generates and tests hypotheses about the unintended visual attributes that a model may rely on. They also introduce a new benchmark of 130 models with synthetically injected, known attribute dependencies, which they use to evaluate their framework.

**Questions:**

See weaknesses.

Minor: What the computational cost (e.g., API calls, time) for analysing a single concept in a model?

**Ethical Concerns:**

["NO or VERY MINOR ethics concerns only"]

**Final Justification:**

Based on the detailed author response and the extra experiments that were supplied and I have raised my score. Below I explain how the rebuttal mitigates my earlier concerns and note the points that are still only partly resolved.

Dependence on the T2I Backbone My principal worry was that bias or drift in the text-to-image (T2I) generator could mislead the agent and yield false positives. The new ablation (random-empty prompts) and the controlled gender-swap attack are helpful. In particular, the report that the agent still recovers the correct reliance in all injected-bias conditions—even if it needs more iterations—suggests a degree of robustness I did not anticipate. That said, the experiments cover just one attribute class (gender) and one corruption type.

Clarification of the Self-Evaluation Protocol The cap-wearing example and the explanation that the system generates two diverse prompt pools, rather than a single minimal pair, remove most of the ambiguity in the original manuscript. The three documented failure-handling behaviors further clarify what happens when generation and text become misaligned.

“Harmful vs. Benign” Reliances The authors now acknowledge that the agent surfaces dependencies without judging their desirability and that task-level judgment is left to practitioners. This reframing should be spelled out early in the paper, especially for readers who might otherwise assume the tool is a bias detector by design.

Outstanding Issues

- The new robustness study is currently limited in scope; expanding to other attributes (e.g., background, material) would strengthen the claim.
- A concise comparison with MAIA’s bias analysis would help further clarify the contribution of the paper.

**Limitations:**

yes

**Quality:**

3

**Strengths And Weaknesses:**

Strengths:

Framing model interpretability as an iterative, scientific discovery process driven by a self-reflective agent is novel and potentially impactful for practitioners.

The introduced benchmark is highly useful for the field and its construction is well described.

Experiments are well designed and the framework is compared to several reasonable benchmarks.


Weaknesses

The framework is dependent on a TTI model to generate images for hypothesis testing and self-evaluation. This introduces a major potential confounder that the paper does not adequately address. TTI models are themselves trained on large web-scale datasets and often use CLIP-like embeddings; thus, they encode their own strong societal and representational biases. For example, when the agent tests a hypothesis like "the model relies on corporate settings for the concept *suit*," it generates images with and without this attribute. However, when it prompts the TTI model for "a person in a suit *not* in a corporate setting," the TTI model might, due to its own biases, change other attributes—it might generate a woman instead of a man, or change the style of the suit. The agent might then observe a drop in the subject model's score and incorrectly conclude that the "corporate setting" was the key attribute, when in fact the reliance was on the person's gender...


The paper frames the discovered dependencies as "unintended" or "vulnerabilities". However, some of the examples seem to be strong, statistically valid features that are part of a concept's representation. For example, the finding that CLIP's vision encoder recognizes "teachers" based on "classroom backgrounds" is not necessarily a spurious shortcut. In many contexts, the background is a defining feature. How else would a model distinguish a teacher from  another person in a professional setting? The framework currently lacks a mechanism to differentiate between a harmful shortcut (e.g., reliance on gender for a job-related concept) and a benign, contextually relevant feature. This makes the practical significance of some findings unclear.

---

> ### Author Rebuttal · Authors · 2025-07-31
>
> We thank the reviewer for their meaningful comments. We were glad to see that, like all other reviewers, reviewer wjA6 found our benchmark and synthetic dataset to be highly useful, and the experiments to be well designed. Below, we address their comments. To give a short summary of the main points:
>
> (1) We clarify the usage of T2I models in the self-evaluation phase and run additional quantitative evaluations to test the agent’s sensitivity to T2I generation failures and biases and find it to be mostly robust to these failure modes. This aligns with the behavior of the agent we observed during experimentation (see description below). We will expand this evaluation and include it in the appendix of the paper.
>
> (2) We clarify the definition of the task and discuss whether all found reliance are harmful—we believe that this depends on the downstream tasks.
> Please see full replies below.
> _____
>
> **T2I models have their own biases**
>
> We agree that T2I models can carry societal and representational biases. This is indeed a general limitation of agentic interpretability methods that rely on T2I models for generating stimuli (and was noted and evaluated in the MAIA paper [1], see their section 4.4).  However, a key distinction in our framework is the inclusion of a self-reflection loop, which we find makes the system more robust to such artifacts.
>
> For example, in the provided supplemental experiment folder “male_cap_0.5”, we provide the full trace of an experiment on a *cap* detection model with an injected bias towards men wearing caps. We observe that the T2I model exhibits gender bias: despite gender-neutral prompts (e.g. “a person wearing a black baseball cap”), all generated images in the first hypothesis-testing round (“round0.html”) depict men wearing caps. As a result, the agent is initially unable to draw any conclusions about potential gender dependencies through hypothesis-testing alone. During self-reflection in round3.html, the agent is evaluating the hypothesis: *“The classifier demonstrates a strong contextual bias toward caps being actively worn on human heads…”* It does so by passing the conclusion to a separate LLM that generates two diverse sets of prompts predicted to elicit high (e.g. *“A young man wearing a red baseball cap while smiling at the camera”*, *“Woman jogger wearing a white sports cap during her morning run”*, *“Teen boy wearing his baseball cap backwards at a skate park”*) and low (e.g. *“New baseball cap displayed on store shelf”*, *“Collection of caps hanging on wall hooks”*, *“Baseball cap floating in swimming pool”*) detection scores, respectively. As a result of the diverse evaluation prompts, the agent is able to recognize a discrepancy in its previous conclusion and formulates a new hypothesis: *“The classifier exhibits a gender and age bias, favoring adult males wearing baseball caps in athletic or casual contexts.”* This insight leads the agent to revise its prompts and explicitly query for images of women wearing caps, ultimately uncovering the correct dependency, despite the T2I model's initial bias.
>
> In several other cases, the agent was able to detect inconsistencies between its generated prompts and the resulting images. Interestingly, when the agent notices such unaligned behavior, it usually uses a different experimental design to get the intended behavior (e.g. if the T2I tool doesn’t follow the prompt correctly, it uses the editing tool the edit one of the dataset exemplars to achieve the desired stimuli) or ignores the incorrect images and focuses its analysis on the successful generations.
>
> To quantify the agent’s robustness to T2I errors, we conducted two experiments:
>
> - *Random empty prompt:* in 50% of T2I calls, we replaced the agent’s prompt with an empty string, resulting in the model generating unrelated content that ignored the intended experimental manipulation. We found that in 80% of these corrupted trials, the agent successfully noticed the issue, either during the hypothesis-testing stage or self-reflection, and responded by revising its approach or ignoring the incorrect images. While this is a simple baseline, it gives an approximation of the agent’s sensitivity to breakdowns in the generation pipeline.
>
> - *Injected bias:* We attack the prompts generated by the agent in experiments on a tie detector system that relies on the presence of a man. We attack the T2I model by systematically replacing instances of the phrase “a person” with “a man” x% of the time for  x = [0, 25, 50, 75, 100], and “a woman” the remaining (100-x)% of the time to simulate a controlled gender bias in the T2I. We found that the agent was robust to attack and able to detect the correct reliance in all simulated bias ratios. Interestingly, the agent required more iterations when the T2I model was biased toward generating images of men (i.e. when the bias of the T2I matched the model’s feature reliance). We noticed this phenomenon in the base T2I model as well, which almost always generates a man when prompted with “a person wearing a tie.” This is another motivating factor for including the study on counterfactual demographic attribute reliant systems (e.g. a tie detector that relies on the presence of a woman)—such systems are out of distribution for not only the multimodal LLM backbone but tools like the T2I as well. This is a first quantitative evaluation that we will expand for the final version of the paper (include more attacks and more systems), and we will report the overall sensitivity of the agent to controlled biases in T2I.
>
>
> [1] Shaham et al. "A multimodal automated interpretability agent." ICML 2024.
>
>
> **Image generation in the self-evaluation might be inaccurate**
>
> To clarify, the self-evaluation protocol does *not* operate by prompting the T2I model with a minimal pair of prompts (e.g. *“a cap on a human head”* vs. *“a cap not on a human head”*). Rather, as described in the previous section, the agent passes its current conclusion (e.g., *“The classifier demonstrates a strong contextual bias toward caps being actively worn on human heads”*) to a separate LLM, which then generates two diverse sets of prompts predicted to elicit high and low detection scores, respectively (e.g. predicted high detection score prompts: *“A young man wearing a red baseball cap while smiling at the camera”*, *“Woman jogger wearing a white sports cap during her morning run”*, *“Teen boy wearing his baseball cap backwards at a skate park”*, and low detection score prompts: *“New baseball cap displayed on store shelf”*, *“Collection of caps hanging on wall hooks”*, *“Baseball cap floating in swimming pool”*). We thank the reviewer for highlighting this unclear description of the self-evaluation protocol and will incorporate a more comprehensive explanation in the revision.
>
>
> We found this strategy to be more robust compared to naive prompting with direct negations (which we indeed found prone to semantic and visual drift). It emphasizes attribute-controlled diversity, where each prompt in the high or low group keeps the core attribute constant (e.g., presence or absence of humans) while allowing other visual factors to vary.
>
>
> When mismatches between textual prompts and generated images is the self evaluation stage do occur, we found three main behaviors:
>
> - The agent is able to detect the misalignment and disregard the artifacted sample (as explained in the previous reply)
>
> - The agent was not able to detect the misalignment, however, the T2I artifact yields an image that doesn’t match the predicted behavior (because of the artifacts), this is typically reflected in a lower predictiveness score. The agent uses this feedback as a signal to reinitiate hypothesis testing (as shown in Fig. 2)
>
> - The agent was not able to detect the misalignment, but the T2I artifact doesn’t affect the related attribute and therefore doesn’t harm the predicted behavior and doesn’t affect the experiments.
>
> We agree that misalignment between textual prompts and generated images in T2I is an important concern. We will add this discussion and the empirical evaluation to the appendix of the paper.
>
> **Found reliance is not necessarily spurious**
>
> We agree. Our framework does not attempt to judge whether a discovered reliance is inherently harmful or contextually appropriate, nor is this distinction communicated to the agent. Rather, the goal is to surface potential dependencies that might not have been previously anticipated, enabling practitioners to make informed judgments based on their specific downstream use case. In some applications, reliance on context (e.g., background for "teacher") may be a valid, even desirable, part of the concept's representation. In others (e.g., medical imaging or fairness-sensitive applications), such dependencies could lead to unwanted generalization or bias. Notably, the agent sometimes concludes that no strong attribute reliance is present. We will make sure to clarify this in the definition of the problem setting in the intro and the rest of the paper.
>
> **Computational cost**
>
> An experiment with 10 hypothesis testing + self-reflection rounds costs approximately $3, and each hypothesis testing + self reflection round takes about 10-20 minutes (Note that most experiments are concluded before 10 rounds, so this is an upper bound).

---

> > ### Author Response · Authors · 2025-08-05
> >
> > We would like to thank the reviewer again for their feedback. As the discussion period is coming to a close, we wanted to check whether our response addresses your concerns. We're happy to provide further clarification if needed.

---

> > ### Comment · Reviewer_wjA6 · 2025-08-05
> > **Thank you for the clarifications, I raised my score**
> >
> > I appreciate the detailed author response and the extra experiments that were supplied and have raised my score. Below I explain how the rebuttal mitigates my earlier concerns and note the points that are still only partly resolved.
> >
> > Dependence on the T2I Backbone
> > My principal worry was that bias or drift in the text-to-image (T2I) generator could mislead the agent and yield false positives. The new ablation (random-empty prompts) and the controlled gender-swap attack are helpful. In particular, the report that the agent still recovers the correct reliance in all injected-bias conditions—even if it needs more iterations—suggests a degree of robustness I did not anticipate. That said, the experiments cover just one attribute class (gender) and one corruption type.
> >
> > Clarification of the Self-Evaluation Protocol
> > The cap-wearing example and the explanation that the system generates two diverse prompt pools, rather than a single minimal pair, remove most of the ambiguity in the original manuscript. The three documented failure-handling behaviors further clarify what happens when generation and text become misaligned.
> >
> > “Harmful vs. Benign” Reliances
> > The authors now acknowledge that the agent surfaces dependencies without judging their desirability and that task-level judgment is left to practitioners. This reframing should be spelled out early in the paper, especially for readers who might otherwise assume the tool is a bias detector by design.
> >
> > Outstanding Issues
> >  - The new robustness study is currently limited in scope; expanding to other attributes (e.g., background, material) would strengthen the claim.
> >  - A concise comparison with MAIA’s bias analysis would help further clarify the contribution of the paper.

---

### Note · Authors · 2025-08-12

We appreciate the time and thoughtful feedback from the reviewers who participated in the discussion and are encouraged that our responses and additional experiments led reviewers wjA6 and aRwo to raise their scores, citing the “detailed author response and the extra experiments” [wjA6] and the “many clarifications and expansions to the synthetic models benchmark, and the experiment showing robustness to failure at the T2I component of the baseline” [aRwo]. We are also grateful to reviewer H1cw for their positive score and feedback (“I find the work interesting and valuable”), and constructive suggestions. All of these reviewers noted that their concerns had been addressed, including robustness to T2I bias and failures, the diversity of prompts generated by the self-evaluation procedure, the complexity of the benchmark models, and generalization to real-world images. These are in addition to the positive points raised in the initial reviews, where all reviewers found our benchmark and synthetic models to be highly useful [wjA6, Cd4M, H1cw, aRwo], the experiments to be well designed [wjA6], recognized self-reflection as a meaningful contribution [Cd4M, H1cw], and the paper to be well written [H1cw, aRwo]. We will ensure that all additional results provided during the discussion are included in the next version of the paper.


Although reviewer Cd4M did not participate in the discussion, we believe we have fully addressed all of their concerns, most of which overlapped with comments from other reviewers who were satisfied with our additional experiments and clarifications. We hope reviewer Cd4M will review our responses and reach the same conclusion.


We hope the AC will take into account the unanimously positive feedback and scores from all reviewers who participated in the discussion when making their decision.

---

### Decision · Program_Chairs · 2025-09-17

**Decision:**

Accept (poster)

**Comment:**

The submission initially received three ratings leaning toward rejection and one recommending acceptance. Reviewers found the proposed self-reflection agent to be novel and interesting, the benchmark useful, the experiments well-designed, and the paper well written. Initial concerns included the reliance on image embeddings or T2I models that may carry inherent biases, the model’s inability to distinguish between harmful and benign attribute dependencies, missing comparison to MAIA, unclear differences between the framework stages, and other technical issues.

The authors rebuttal effectively addressed most of these concerns, and all reviewers shifted to recommending acceptance in their final ratings.

The AC concurs with the reviewers consensus and recommends acceptance. The AC asks the authors to incorporate the rebuttal and reviewers feedback into the final version.